# Dissociation of impulsive traits by subthalamic metabotropic glutamate receptor 4

**Lukasz Piszczek[1†], Andreea Constantinescu[1†], Dominic Kargl[1,2†], Jelena Lazovic[3], Anton Pekcec[4], Janet R Nicholson[4], Wulf Haubensak[1,2]\***

[1]The Research Institute of Molecular Pathology (IMP), Department of Neuroscience, Vienna Biocenter, Vienna, Austria; [2]Department of Neuronal Cell Biology, Center for Brain Research, Medical University of Vienna, Vienna, Austria; [3]Preclinical Imaging Facility, Vienna BioCenter Core Facilities (VBCF), Vienna, Austria; [4]Div Research Germany, Boehringer Ingelheim, Biberach an der Riss, Germany

**Abstract** Behavioral strategies require gating of premature responses to optimize outcomes. Several brain areas control impulsive actions, but the neuronal basis of natural variation in impulsivity between individuals remains largely unknown. Here, by combining a Go/No-Go behavioral assay with resting-state (rs) functional MRI in mice, we identified the subthalamic nucleus (STN), a known gate for motor control in the basal ganglia, as a major hotspot for trait impulsivity. In vivo recorded STN neural activity encoded impulsive action as a separable state from basic motor control, characterized by decoupled STN/substantia nigra pars reticulata (SNr) mesoscale networks. Optogenetic modulation of STN activity bidirectionally controlled impulsive behavior. Pharmacological and genetic manipulations showed that these impulsive actions are modulated by metabotropic glutamate receptor 4 (mGlu4) function in STN and its coupling to SNr in a behavioral trait-dependent manner, and independently of general motor function. In conclusion, STN circuitry multiplexes motor control and trait impulsivity, which are molecularly dissociated by mGlu4. This provides a potential mechanism for the genetic modulation of impulsive behavior, a clinically relevant predictor for developing psychiatric disorders associated with impulsivity.

**\*For correspondence:**
wulf.haubensak@imp.ac.at

[†]These authors contributed equally to this work

## Editor's evaluation

Piszczek et al., probed mGluR4 modulation of impulsivity at the systems and molecular level using resting fMRI, in vivo/ex vivo electrophysiology, pharmacological, optogenetic, and genetic manipulations in mice. Authors link behavioral trait variation to an mGluR4-dependent mechanism in the subthalamic nucleus. Overall, the identification of mGlu4 as a marker for high trait impulsivity reveals a novel potential therapeutic.

## Introduction

Successful environmental interactions require fast, but conditional execution of appropriate behavioral responses. Impulse control adapts the balance between action and action inhibition to optimize behavioral outcomes. The past decade has seen a wealth of advances in our knowledge of the neuronal basis of impulsive action. Functional magnetic resonance imaging (fMRI) studies in humans and behavioral studies in animal models (*Eagle et al., 2008a*; *Winstanley et al., 2006*) have delineated key areas in the brain that control impulsive action (*Bari and Robbins, 2013*; *Dalley and Robbins, 2017*). Several interconnected brain areas, including the prefrontal cortex (*Beauchaine et al., 2017*;

*Feja and Koch, 2015*; *Huang et al., 2017*; *McDonald et al., 2017*), anterior cingulate cortex (*Huang et al., 2017*; *Kerr et al., 2015*), insular cortex (*Belin-Rauscent et al., 2016*; *Ishii et al., 2012*; *Swick et al., 2011*), amygdala (*Kerr et al., 2015*; *Xie et al., 2011*; *Zeeb and Winstanley, 2011*), subthalamic nucleus (STN) (*Bari and Robbins, 2013*; *Jahanshahi et al., 2015*; *Uslaner and Robinson, 2006*; *van Hulst et al., 2017*; *Yoon et al., 2019*), and nucleus accumbens (*Cardinal et al., 2001*; *Costa Dias et al., 2013*; *Economidou et al., 2012*), among others (*Simmonds et al., 2008*), control various aspects of impulsive behavior. These regions process multiple brain functions related to impulsivity, such as reward processing, decision-making, and motor execution (*Dalley and Robbins, 2017*), in addition to their canonical functions. Consequently, they encode behavioral parameters of impulsive choices as discrete features within broader patterns of neuronal activity (*Dalley and Robbins, 2017*; *Schmidt et al., 2013*). Genetically, many genes have been associated with distinct forms of impulsivity and impulsivity-related disorders, such as attention deficit hyperactivity disorder (ADHD) (*Bralten, 2013*; *Lasky-Su et al., 2008*; *Naaijen, 2017*), which mainly affect serotonergic (*Harrison et al., 1999*) and dopaminergic systems (*Jupp et al., 2013*; *Simon et al., 2013*) and their glutamatergic modulation (*Isherwood, 2017*).

As with many cognitive functions, impulsive actions underlie a natural variance expressed as a specific behavioral trait in both humans (*Angelides et al., 2017*; *Ding et al., 2014*; *Huang et al., 2017*) and rodents (*Gubner et al., 2010*; *Isles et al., 2004*; *Loos et al., 2015*; *Loos et al., 2010*). This trait impulsivity can be defined as a preference for immediate rewards over larger delayed rewards. In this regard, moderate trait impulsivity involves taking calculated risks or pursuing unexpected outcomes to maximize overall gain or rewards. Excessive trait impulsivity involves unreasonably risky, premature behaviors resulting in negative consequences, such as punishments (*Dalley and Robbins, 2017*). According to the manual for assessment and diagnosis of mental disorders, DSM-V, pathological trait impulsivity is a key diagnostic indicator of impulse control disorders (ICDs), which have attracted significant public health interest in recent years. Moreover, excessive impulsivity contributes significantly to the pathology of mood disorders, drug abuse, and addiction, as well as ADHD and borderline personality disorder (*Beauchaine et al., 2017*; *Dougherty et al., 2004*; *Ersche et al., 2010*; *Lombardo et al., 2012*; *Perry and Körner, 2011*; *Perry and Carroll, 2008*; *Rentrop et al., 2008*). Understanding the neuronal basis of trait impulsivity, thus, is key to identifying risk factors and progression toward these conditions.

There has been substantial progress in characterizing the diverse neuronal circuitry and mechanisms that control impulsive action per se, but we know much less about the neuronal basis underlying natural variation in impulsivity. This manifests as stably expressed individual behavioral traits, programmed within impulsive circuitry (above), and separated from other brain functions. At the molecular level, the expression of a behavioral trait is shaped by genetic/epigenetic factors (*Loos et al., 2009*). The mechanisms that link individual genes and epigenetic factors to specific modulation of neuronal activity and trait impulsivity are poorly understood, however. To address these mechanisms, we used a free-moving, Go/No-Go (GNG) task (*Gubner et al., 2010*; *Harrison et al., 1999*; *Humby and Wilkinson, 2011*) to study the natural variance in experimental cohorts. This task involves a cued preparatory phase followed by two cues, to which the subjects must either respond rapidly or withhold a response (*Hong et al., 2016*; *Simmonds et al., 2008*) in order to gain a reward. This allowed us to systematically chart hotspots for trait impulsivity and investigate their genetic modulation. This strategy identified the STN as a site where metabotropic glutamate attenuates neural activity and impulsive choices, without affecting general motor output. From a translational perspective, this promised insight into biomedically relevant mechanisms underlying trait impulsivity. More generally, this study explored how modulatory genes may dissociate specific cognitive traits from other brain functions multiplexed within the same circuitry.

## Results

### Brain-wide rs-fMRI associates STN with trait impulsivity

To screen for brain circuitry underlying trait impulsivity, we modeled variant impulsive behaviors in isogenic strains as proxy (*Isles et al., 2004*; *Loos et al., 2015*; *Loos et al., 2010*). Compared to more complex scenarios addressed by between strain comparisons, this strategy was designed to identify basic functional differences emerging from a single genetic and neuroanatomical context,

here C57BL/6. We chose a free-moving variant of a GNG task (*Gubner et al., 2010*) with signaled trials. This assay allowed us to monitor a broad range of behavioral characteristics within a given task session, from general motor behavior to several impulsivity-related parameters, which was not possible with head-fixed or port-fixed variants of this task (*Allen et al., 2017*; *Bathellier et al., 2012*; *Berdichevskaia and Cazé, 2016*; *Montijn et al., 2015*). Each trial of the task was automatically initiated by a light cue that signaled the beginning of the precue period (*Figure 1A*). Responses during this period were recorded (precue response rate), but neither rewarded nor penalized. After a randomized time period, an auditory cue was presented signaling either a Go or a No-Go (NG) trial. Correct reresponses to both Go (response) and NG (withhold) cues were rewarded. Conversely, incorrect Go and NG trials were noted as omission and false alarm (FA), respectively, and no reward was delivered (see Materials and methods for details). Precue response rate and FAs provide two important but distinct parameters of impulsivity in this task. FA responses in the port during the presentation of the NG cue indicate the capacity to restrain prepotent motor responses (called stopping impulsivity), indicating cognitive executions; they are thus an index of failure in cue-related withhold behavior. Precue responses, by contrast, reflect impulsive action in the preparatory phase of the task, indicating how long an animal is willing to withhold a response (called waiting impulsivity) before a cue appears (*Dalley and Robbins, 2017*; *Gubner et al., 2010*; *Moschak et al., 2013*; *Moschak et al., 2012*; *Moschak and Mitchell, 2012*). These forms of impulsivity may have different translational implications for psychiatric symptoms, as their involvement in pathobiology for obsessive compulsive and addiction-related conditions, respectively, has been shown (*Broos et al., 2012*; *Eagle et al., 2008a*).

To map hotspots in trait impulsivity within brain networks, we first trained an initial cohort of mice in our cued GNG task (*Figure 1A*). We then set a behavioral contrast between the top 75th percentiles and bottom 25th percentiles on a compound impulsivity measure of both precue response rate and incorrect NG responses (FA) (*Figure 1—figure supplement 1A*). This served as a boundary for high and low impulsive (HI and LI) animals, respectively (*Figure 1B*), which expressed stable impulsive traits over the course of at least three sessions (*Figure 1Ci*,ii). To isolate variance in impulsive behavior, independent of motivational state or performance in task execution, HI/LI groups were filtered for similar levels of correct Go responses (>95%, *Figure 1Ciii* and *Figure 1—figure supplement 1B*) and total number of licks at reward delivery (*Figure 1—figure supplement 1C*).

Both the HI and LI groups were analyzed by resting-state functional magnetic resonance imaging (rs-fMRI), as straightforward means to map differences in brain network connectivity – in this case, differences that correlate with HI/LI behavioral traits. The goal of the fMRI analysis was, first and foremost, to highlight the top ranked nodes for impulsive traits. To establish such a ranked list reliably, we used an experimental design suitable for small sample sizes (*Egimendia et al., 2019*; *Sirmpilatze et al., 2019*) to limit the number of animals subjected to behavioral handling and the fMRI operating time and associated costs. Such a statistically low-powered screen is liable to false negatives; however, this design and the following analyses were tailored to identify the top-most ranked nodes. To trace the most prominent changes in the general brain network, we screened seed-wise for the most affected nodes. To this end, we rank-ordered node connectivity differences between HI and LI groups using group means to reduce effects from individual animal variance in low sample size settings (*Bero, 2012*; *Cruces-Solis et al., 2020*; *Filipello et al., 2018*; *Hsu et al., 2012*; *Liu et al., 2020*; *Rosenberg et al., 2016*; *Tsurugizawa et al., 2020*). Specifically, after computing the brain-wide functional connectivity in HI and LI animals for each region (node), we determined differences in connectivity between the two groups, which were then evaluated for statistical significance. For the comparison of HI and LI groups, these subtracted correlations report effect sizes and rank more intuitively than simple p-values, in particular given the low sample size. We note that this data could be extended to delineate the associated specific impulsivity functional subnetworks (edges) in a follow-up graph theoretical workup of our data.

Our rs-fMRI analysis was designed to locate hotspots (i.e., nodes) related to impulsive traits. This analysis revealed 13 (of 51) brain nodes that differed in their functional connectivity between HI and LI groups (*Figure 1D*, *Figure 1—figure supplement 2A-Ci*). In HI animals, several brain nodes had higher functional connectivity, when compared to their LI littermates; these included the superior central raphe nucleus (CSm), midbrain reticular nucleus (MRN), periaqueductal gray (PAG), agranular insular area (AI), infralimbic cortex (ILA), and nucleus accumbens (ACB). Previous studies have implicated many of these areas in modulating impulsive behavior, goal-directed actions, and in responding

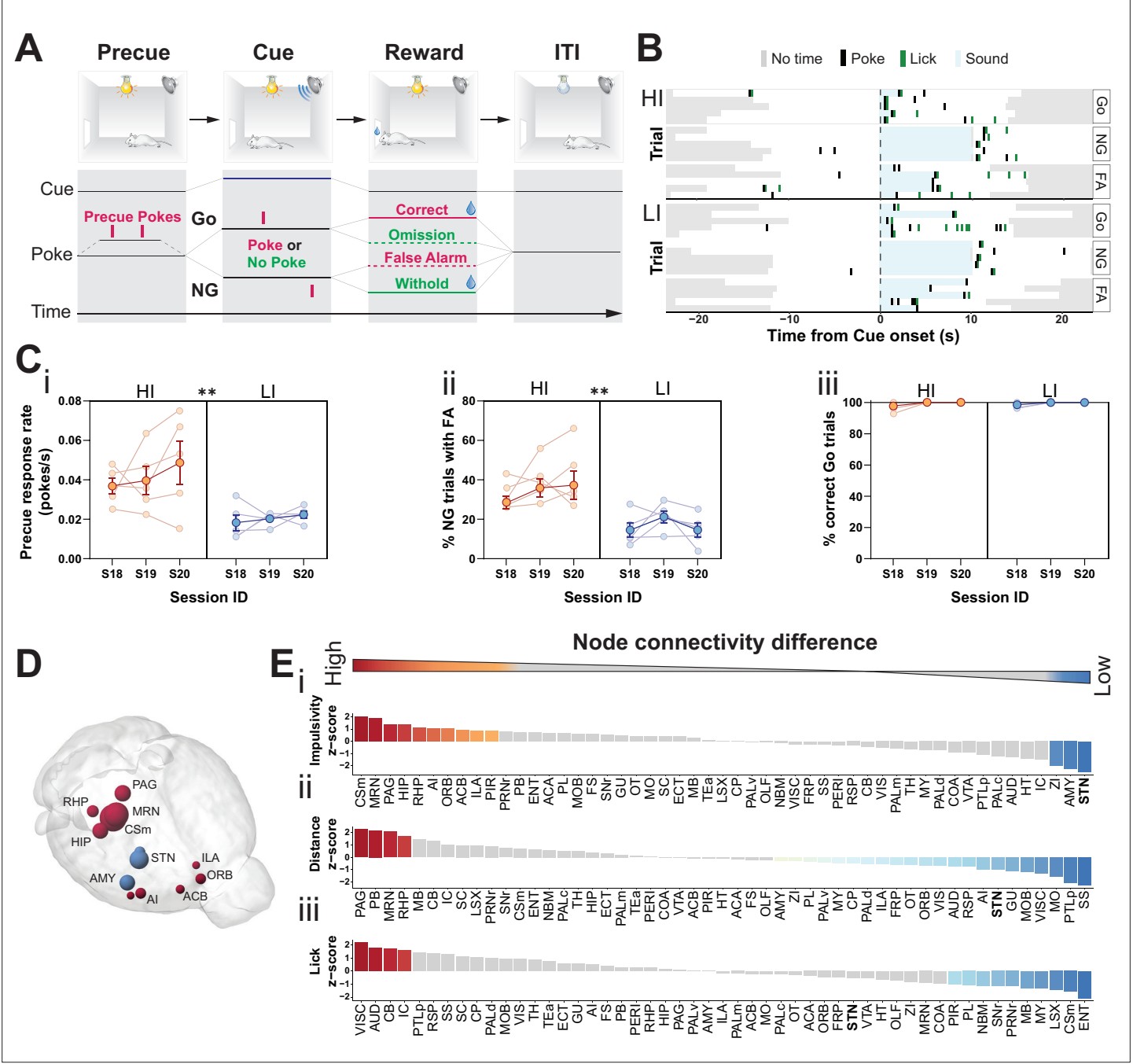

**Figure 1.** Brain-wide resting-state functional magnetic resonance imaging (rs-fMRI) associates subthalamic nucleus (STN) with trait impulsivity. (**A**) Schematic description of the Go/No-Go (GNG) task. The start of each trial is cued by the light turning on (precue period). Precue responses are recorded. Correct Go responses and No-Go (NG) withholds are rewarded. Incorrect Go and NG responses are recorded as an omission and false alarm (FA), respectively, with no reward delivery (see Materials and methods for details). (**B**) Example of behavioral recording of individual animals from either the high impulsive (HI) or low impulsive (LI) group. Each row represents a single trial. Correct responses to a Go (Go) or NG trial, and FA responses to a NG trial (FA) are grouped. (**C**) Behavioral split into HI (N = 5) and LI (N = 5) animals (see *Figure 1—figure supplement 1A*) showing stable differences in (**i**) precue response rate (main effect of impulsivity $F_{1,8}$ = 11.69, p = 0.009, no main effect of session $F_{2,16}$ = 1.454, p = 0.263 and no interaction $F_{2,16}$ = 0.389, p = 0.684). (**ii**) FAs (main effect of impulsivity $F_{1,8}$ = 17.82, p = 0.003, no effect of session $F_{2,16}$ = 1.511, p = 0.251 and no interaction $F_{2,16}$ = 0.710, p = 0.507) in three consecutive sessions (S18–S20). (**iii**) A significant main effect for session was found for % of correct Go trials ($F_{2,16}$ = 4.558, p = 0.027), but no main effect of impulsivity ($F_{2,16}$ = 0.207, p = 0.815) or interaction ($F_{2,16}$ = 0.207, p = 0.815). Group means ± SEM and single animal data in the background in the last three training sessions. (**D, E**) Brain-wide rs-fMRI identifies differences in functional connectivity. (**D**) 3D visualization of the node-wise one-sample t-tests from node-wise connectivity difference between HI vs. LI showing the STN as the node with the highest functional connectivity in LI animals when

*Figure 1 continued on next page*

*Figure 1 continued*

compared to HI. Color indicates that the mean node connectivity is higher in the HI group (red) or in LI animals (blue); size correlates with t-value. Only significantly different nodes are shown. (**E**) Ordered, normalized one-sample t-test t-values from node-wise connectivity strength measurements for splits based on (**i**) impulsivity parameters, (**ii**) total distance traveled, and (**iii**) number of licks in the GNG task. Gray bars indicate that the p value did not reach significance. Bonferroni corrected for multiple comparisons. *p < 0.05, **p < 0.01, ***p < 0.001, ****p < 0.0001. ACB – nucleus accumbens; AI – agranular insular area; AMY – amygdalar nuclei; Csm – superior central nucleus raphe; HIP – hippocampus; ILA – infralimbic area; MRN – MB reticular nucleus; ORB – orbital area; PAG – periaqueductal gray; RHP – retrohippocampal region; STN – subthalamic nucleus.

The online version of this article includes the following source data and figure supplement(s) for figure 1:

**Source data 1.** Related to *Figure 1C and E*.

**Figure supplement 1.** Behavioral splits of animals for the resting-state functional magnetic resonance imaging (rs-fMRI) screen.

**Figure supplement 1—source data 1.** Related to *Figure 1—figure supplement 1A–D*.

**Figure supplement 2.** Brain-wide resting-state functional magnetic resonance imaging (rs-fMRI) screen for high impulsive (HI) vs. low impulsive (LI) animals.

**Figure supplement 2—source data 1.** Related to *Figure 1—figure supplement 2C*.

to rewards (*Baldo et al., 2016*; *Belin-Rauscent et al., 2016*; *Clark et al., 2008*; *Dalley et al., 2007*; *Eagle et al., 2009*; *Harrison et al., 1999*; *Homberg, 2007*; *Ishii et al., 2012*; *Lê et al., 2008*; *Liu et al., 2004*; *Parkes et al., 2015*; *Sesia et al., 2010*; *Sesia et al., 2008*; *Swick et al., 2011*). In LI animals, by contrast, the STN had the highest functional connectivity when compared to the HI littermates, followed by the zona incerta (ZI) and amygdala (AMY) (*Figure 1D–Ei*, *Figure 1—figure supplement 2A,Ci*). Among these regions, the STN is a key region embedded in the basal ganglia known as the 'indirect pathway of movement'. Thus, it is ideally situated to control motor activity in impulsivity-related tasks in animals (*Eagle et al., 2008b*; *Eagle et al., 2008a*; *Schweizer et al., 2014*; *Wiener et al., 2008*) and humans (*Aron and Poldrack, 2006*; *Bastin, 2014*; *Herz et al., 2014*; *Whelan, 2012*; *Yoon et al., 2019*; *Zavala et al., 2018*; *Zavala et al., 2017*).

To contrast functional networks for trait impulsivity with those from other behavioral parameters, we analyzed the rs-fMRI data of our cohort for brain regions involved in other behavioral parameters: the total distance traveled in the task (*Figure 1Eii*, *Figure 1—figure supplement 1Di*, *Figure 1—figure supplement 2Cii*) and the total number of licks during the task, which indicates reward collection and is a proxy for the motivational state of the animals (*Figure 1Eiii*, *Figure 1—figure supplement 1Dii*, *Figure 1—figure supplement 2Ciii*). Analysis of the effect sizes showed that STN ranked top for trait impulsivity but scored considerably lower for motor drive (*Figure 1E* and *Figure 1—figure supplement 2C*); the ZI and AMY, likewise, ranked high for trait impulsivity and low for motor drive. The MRN and PAG, by contrast, showed strong differences for both impulsivity and distance traveled, whereas they showed no significant effect for licks. The CSm showed a high correlation for HI parameters but the opposite effect for licks. Lastly, we found no brain region significantly different in all three parameters.

Taken together, the rs-fMRI screen identified the STN as a modulatory hub with comparably high specificity for trait impulsivity. This suggests that the STN, in addition to its canonical role in basic motor control, is also the main site of naturally occurring variance in trait impulsivity.

## STN differentially encodes impulsive features and motor states

To dissect further the specific roles of the STN in impulsivity and motor function, we performed in vivo extracellular recordings during the GNG task (*Figure 2—figure supplement 1*, *Figure 2—figure supplement 2A-B*). We found mild responses in the STN in response to the light cue that signaled the beginning of the precue period (*Figure 2Ai*), indicating specific activity in the waiting period of the task. As expected, we also found responses to the Go sound onset (*Figure 2Aii*), but we saw no significant response to the auditory cue signaling an NG trial, whether the animals responded with an FA (*Figure 2Aiii*) or with a correct withhold (*Figure 2—figure supplement 2C*). Consistent with the role of the STN in reward processing (*Breysse et al., 2015*; *Espinosa-Parrilla et al., 2013*; *Lardeux et al., 2009*), several units modulated their activity upon reward (*Figure 2Aiv*). Surprisingly, reward-related information is processed by units separate from those potentially gating Go responses (*Figure 2—figure supplement 2B*, *Figure 2—figure supplement 2E*). From these data, we conclude that responses to the Go cue and to the reward are encoded by separate channels in

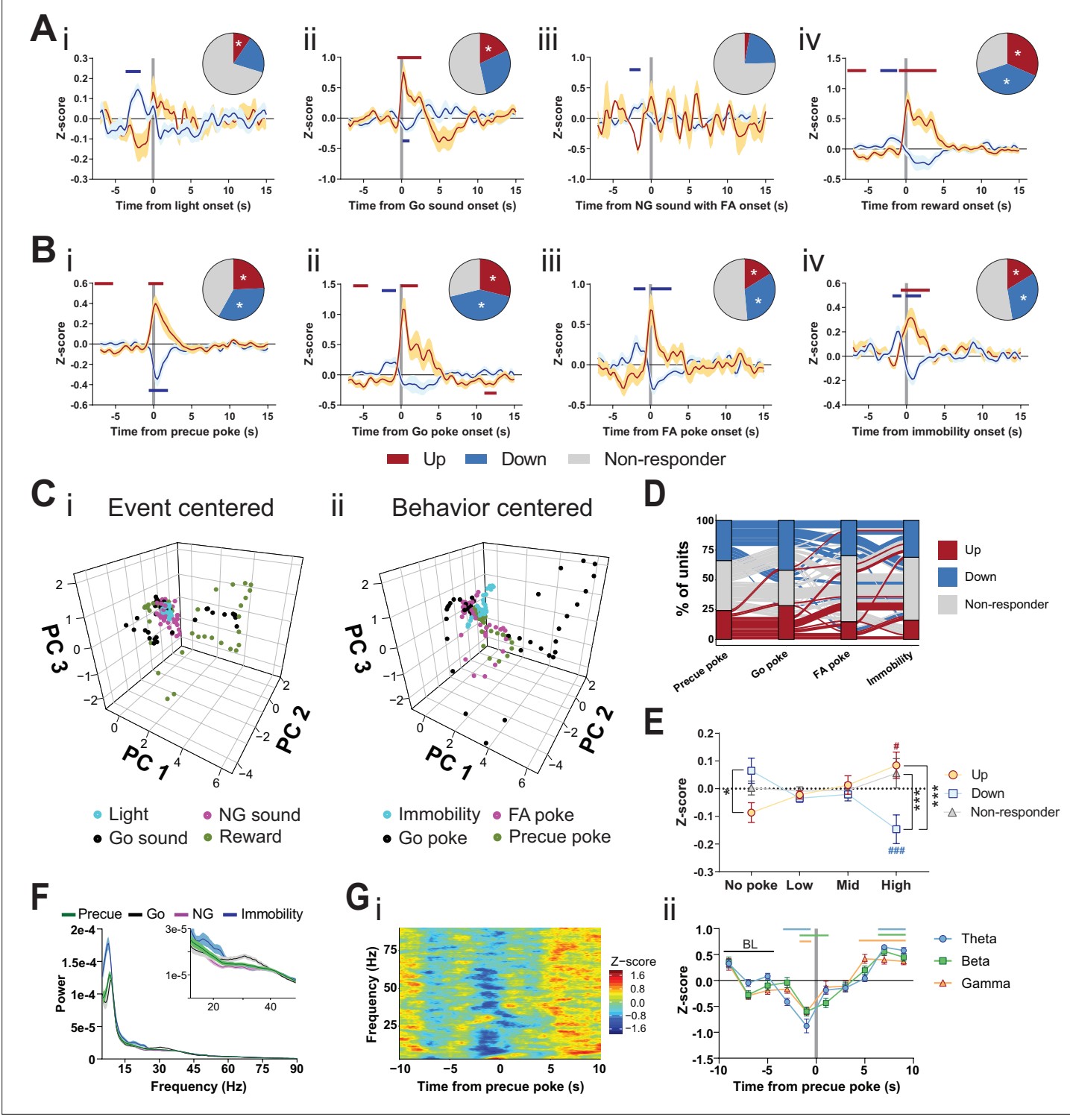

**Figure 2.** Subthalamic nucleus (STN) differentiates impulsivity from motor parameters in the Go/No-Go (GNG) task. (**A, B**) Population traces of excited (up) and inhibited (down) units. (**A**) Peri-event data of unit responses aligned to stimulus onsets: light onset (**Ai**, $n_{Up}$ = 7, $n_{Down}$ = 15 from $N_{Animals}$ = 3), Go sound followed by correct response (**Aii**), $n_{Up}$ = 13, $n_{Down}$ = 21 from $N_{Animals}$ = 3), No-Go (NG) sound followed by false alarm (FA) (**Aiii**) $n_{Up}$ = 3, $n_{Down}$ = 14 from $N_{Animals}$ = 3), reward (**Aiv**, $n_{Up}$ = 23, $n_{Down}$ = 28 from $N_{Animals}$ = 3) and behavioral onsets (**B**) of precue poke (**Bi**, $n_{Up}$ = 18, $n_{Down}$ = 25), Go poke (**Bii**, $n_{Up}$ = 21, $n_{Down}$ = 31 from $N_{Animals}$ = 3), FA poke (**Biii**, $n_{Up}$ = 11, $n_{Down}$ = 22 from $N_{Animals}$ = 3), immobility (**Biv**, $n_{Up}$ = 12, $n_{Down}$ = 23 from $N_{Animals}$ = 3). Horizontal colored bars at the top of each panel indicate time bins with a significantly different firing rate to the baseline firing rate ($p < 0.05$, cluster-based permutation tests). Circles represent the proportion of cells excited (red), inhibited (blue), or non-responsive to a given event (* given population reached significance in the permutation test). (**C**) Population activity vectors in PCA space for stimulus-driven peri-events in panel (**A**) (**i**) and behavior-

*Figure 2 continued on next page*

*Figure 2 continued*

driven peri-events shown in panel (**B**) (ii) in a –5 to 5 s time window. (**D**) Alluvial plot of individual STN units to precue poke, Go sound with correct response, NG sound followed by FA and immobility onsets. The width of the ribbon is proportional to the fraction of units with a given response pattern. Ribbons are color coded according to precue poke responses. (**E**) Change in mean firing rate during the entire precue period of unit populations (split by response to precue poke) on trials with no precue poke, low (<0.15), mid (0.15–0.3), and high (>0.3) precue response rate ($n_{Up}$ = 18, $n_{Down}$ = 25, $n_{Non-responder}$ = 31 from $N_{Animals}$ = 3). *p < 0.05, ***p < 0.001. (**F**) LFP power spectra in STN during precue, Go sound, NG sound, and immobility periods ($N_{animals}$ = 3). (**G**) Within frequency Z-scored LFP spectrogram (**i**) and frequency band-averaged, 2s-binned time course thereof (**ii**) in the STN, both centered around precue pokes. Mean Z-score ± SEM from $n_{Channels}$ = 48, $N_{Animals}$ = 3. Colored lines indicate significant differences to the baseline (BL) period in the respective frequency band, as determined by two-way ANOVA (significant main effect of time ($F_{4.438,625.8}$ = 65.59, p < 0.0001) and time × frequency band interaction ($F_{14,987}$ = 2.926, p = 0.0002) and Dunnet post hoc analysis (p < 0.05)).

The online version of this article includes the following source data and figure supplement(s) for figure 2:

**Source data 1.** Related to *Figure 2A, B, E, F and Gii*.

**Figure supplement 1.** Electrode placement.

**Figure supplement 2.** Encoding of Go/No-Go (GNG) task parameters in the subthalamic nucleus (STN).

**Figure supplement 2—source data 1.** Related to *Figure 2—figure supplement 2B, C, D*,F.

**Figure supplement 3.** LFP time courses for Go/No-Go (GNG) task parameters in the subthalamic nucleus (STN).

**Figure supplement 3—source data 1.** Related to *Figure 2—figure supplement 3*.

the STN. Using distinct channels may uncouple reward value from response control during conditional responding. Our analysis also revealed STN responses coupled to the precue, Go, and FA pokes (*Figure 2Bi-iii*), and we detected unit populations within the STN that responded to the onset of immobility (*Figure 2Biv*) and movement (*Figure 2—figure supplement 2D*). These immobility onsets were spread across various task periods; the majority occurred either in the precue or ITI periods of the task, and fewer than 5% occurred during presentation of the NG sound (*Figure 2—figure supplement 2F*). These data indicate that the STN encodes preparatory withholding, akin to behavioral inhibition, rather than cognitive control of Go vs. NG cue-dependent behavioral discrimination (*Gubner et al., 2010*).

We next explored the encoding of these features in the principal component analysis (PCA) space of the population vector activity in the STN. We found distinct vector paths for subsets of these parameters, which suggests rather rigid separation of some stimulus and behavior variables. Responses to the Go sound and reward onset had similar trajectories in PCA space (*Figure 2Ci*), with weak general response to both light and NG sound onsets. Moreover, cell-by-cell analysis revealed that these states were encoded across STN units and each unit carried multiple signals (*Figure 2—figure supplement 2E*). Precue poke trajectories (and to a lesser extent FA pokes) were separated from the trajectories for immobility and Go pokes (*Figure 2Cii*, *Figure 2—figure supplement 2G*). These data indicate that the STN encodes features of impulsivity, particularly those associated with precue pokes, as a behavioral state distinct and dissociable from Go responses. The sets of units that responded to precue pokes with decreased firing rates, varied in their firing patterns for other behavioral variables, for example, non-responding, increasing, or decreasing upon Go poke or immobility onsets (*Figure 2D*). Taken together, these findings suggest multiplexed encoding of impulsivity and motor features in the GNG task at the unit level.

Since many of the investigated units were bound to a precue poke event, we examined whether there might be a correlation between impulsivity behavioral events in this task period and STN electrophysiological activity. As a proxy for an animal split on impulsivity level, we grouped the pooled precue periods into four categories with increasing number of behavioral impulsivity events – the precue pokes (*Figure 2E*). The unit population suppressed during the precue poke event significantly decreased its average firing rate with increased incidence of behavioral events (negatively correlated with precue pokes). However, the activated population showed the converse effect (positively correlated with precue pokes). Importantly, the mean activity of the negatively correlated population was higher in periods without precue pokes compared with units activated by this behavioral event, strongly suggesting that this population acts as an inhibitory gatekeeper for this behavior.

Local units are bound by time-locked oscillations that organize functional coupling to intra-STN and mesoscale brain networks. Specifically, beta-range coupling in the STN is associated with action inhibition (*Leventhal, 2012*; *Schmidt et al., 2013*), whereas gamma-range coupling is associated

with action execution (*Jenkinson et al., 2013*). To study the oscillatory activity during the GNG task, we compared event- (precue, Go and NG sound periods) and behavior- (immobility) related spectral powers in the task (*Figure 2F*). High theta power during immobility might reflect pre-decision-making for action selection (*Heikenfeld et al., 2020*; *Zavala et al., 2015*). As expected from the role of STN in action inhibition, the immobility period showed the strongest LFP power in the beta band (*Figure 2F* inset), while the highest gamma power was bound to the Go sound cue (*Figure 2F* inset), reflecting beta- and gamma-related behavioral inhibition and execution, respectively. To investigate these patterns in more detail, we used time-resolved power spectrograms centered around the onset of events and behaviors. This analysis revealed action-locked STN activity probably linked to reward expectation/consumption (*Figure 2—figure supplement 3Bii/Ai*). This signal was absent from NG-cued and non-reinforced FA pokes (*Figure 2—figure supplement 2Aii*). By contrast, immobility onset correlated with an overall increase in theta power, followed by a drop of beta and gamma activity (*Figure 2—figure supplement 3Aiii*). Unlike the ITI periods, trial onsets significantly reduced theta and beta bands, suggesting behavioral disinhibition and lower action thresholds during the precue period (*Figure 2—figure supplement 3Bi*). This pattern was less pronounced during the NG sound, indicating a bias toward waiting impulsivity in the STN (*Figure 2—figure supplement 3Biii*). Consequently, by analyzing the power spectra around precue pokes, we found a strong decrease in the theta, beta, and gamma bands prior to a precue poke (*Figure 2G*), suggesting that neuronal decoupling of STN from its intra-STN and meso-scale networks results in impulsive choice. Overall, this picture recapitulates the decoupling of the STN from global networks in HI animals (*Figure 1Ei*).

## Optogenetic perturbation of the STN modulates impulsivity

To assess the gatekeeping function of the STN in the GNG task, we used an optogenetic approach. By using adeno-associated viral vectors, we expressed the light-sensitive opsins channelrhodopsin-2 (ChR2) or archaerhodopsin (Arch) in the STN and implanted an optical fiber above the STN for light activation (*Figure 3—figure supplement 1*). Prior to behavioral testing, we functionally validated this approach by patch clamp recordings to measure activation and inhibition of the STN (*Figure 3—figure supplement 2A*).

The temporal resolution of optogenetics allowed us to investigate STN function specifically in the precue and cue phases of the task. Inhibition of the STN during the precue phase significantly increased the precue response rate, whereas activation had the opposite trend (*Figure 3A and Bi*, *Figure 3—figure supplement 2Bi*). These opposite effects strongly suggest bidirectional control of impulsivity by the STN. Under this stimulation regime, we also observed post-laser effects: the latency to respond to both Go (*Figure 3—figure supplement 2Bii*) and NG (*Figure 3—figure supplement 2Cii*) cues was prolonged in the manipulations, however, it did not strongly impact the number of responses in either the Go or NG trials (*Figure 3Bii*, *Figure 3—figure supplement 2Ci*).

As expected, cue-bound optogenetic manipulation of STN function did not affect the precue response rate (*Figure 3C and Dii*), however, it did reduce the fraction of Go responses in both the ChR2 and Arch groups (*Figure 3Di*, *Figure 3—figure supplement 2Di*), with alterations in latency to respond (*Figure 3—figure supplement 2Dii*). We speculate that this unidirectional effect indicates that activation or inhibition of the STN impacts general task performance (as measured by the Go parameter). Consistent with this, STN lesions in animals and deep brain stimulation (DBS) in humans altered the number of cue responses (*Baunez and Robbins, 1997*; *Hershey et al., 2010*). Also, studies in which STN activity was increased by blocking GABA-ergic inputs resulted in abnormal movements and decreased behavioral performance (*Karachi et al., 2009*; *Périer et al., 2002*). Optogenetic activation or inhibition of STN function had no effect in NG trials on either response numbers (*Figure 3—figure supplement 2Ei*) or their latency (*Figure 3—figure supplement 2Eii*), thus dissociating precue effects on waiting from cue effects stopping impulsivity. This suggests that, at lower intensity regimes (see below, *Figure 6—figure supplement 4A*), STN function mainly affects waiting impulsivity and, to a lesser extent, cue-related responses, even though we found precue responses and FA responses were both represented in the STN (*Figure 2Cii*, FA-related trajectories).

Taken together, our findings from optogenetics and electrophysiological recordings show that the STN encodes and controls impulsive action, especially related to precue waiting impulsivity.

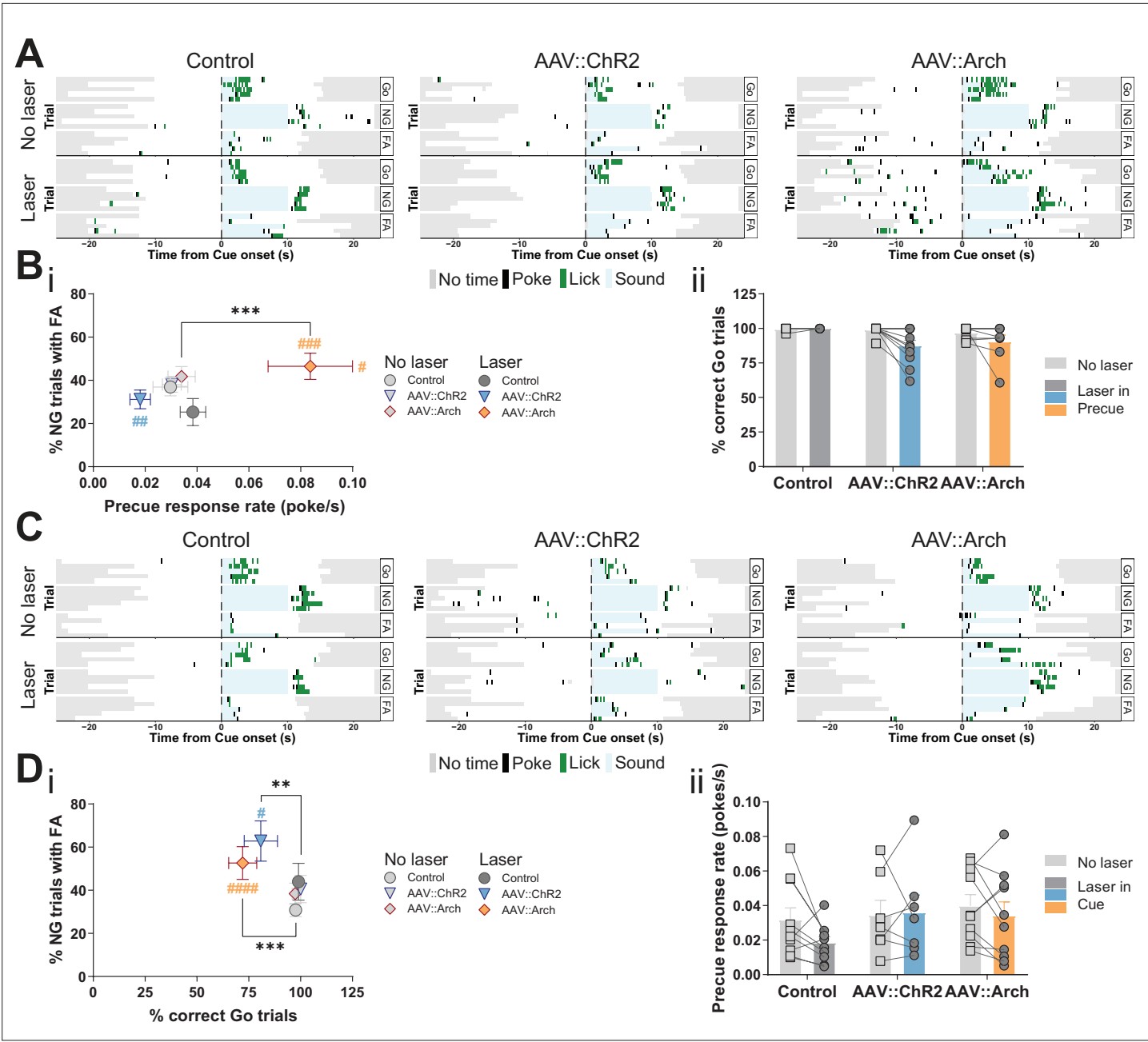

**Figure 3.** Optogenetic perturbation of the STN modulates impulsivity. Manipulation of STN activity during the precue period (**A–B**) or cue period (**C–D**) in the Go/No-Go (GNG) task. (**A, C**) Example behavioral traces for single animals from control (AAV::GFP), STN activation (AAV::ChR2), and STN inhibition (AAV::Arch) groups in the correct Go (Go), correct No-Go (NG), and NG trials with false alarm (FA). Top shows a no laser session. Bottom shows a behavior session with laser during the respective task period. Each row represents a single trial. (**B**) STN manipulation during the precue period (**i**) affects impulsivity parameters (two-way repeated measures (RM) MANOVA main effect of laser time $F_{1,22} = 5.921$, $p = 0.0087$, main effect of virus group $F_{2,46} = 4.691$, $p = 0.0029$, and interaction between the two $F_{2,46} = 4.821$, $p = 0.0025$). Two-way RM ANOVA on % correct Go trials (ii) showed a significant main effect of laser time ($F_{1,23} = 6.985$, $p = 0.0145$), but no significant main effect of virus group ($F_{2,23} = 2.584$, $p = 0.097$) and no significant interaction between the two ($F_{2,23} = 2.613$, $p = 0.095$). $N_{Control} = 7$, $N_{AAV::ChR2} = 11$ and $N_{AAV::Arch} = 8$. (**D**) Manipulation of the STN during the cue period (**i**) affects GNG task parameters (two-way RM MANOVA main effect of laser time $F_{1,23} = 23.52$, $p < 0.0001$, main effect of virus group $F_{2,48} = 4.309$, $p = 0.0029$, and interaction between the two $F_{2,48} = 3.673$, $p = 0.0109$). Two-way RM ANOVA on precue response rate (ii) showed no significant main effect of laser time ($F_{1,24} = 2.089$, $p = 0.161$), no significant main effect of virus group ($F_{2,24} = 1.08$, $p = 0.356$) and no interaction between the two ($F_{2,24} = 1.1$, $p = 0.349$). $N_{Control} = 10$, $N_{AAV::ChR2} = 7$ and $N_{AAV::Arch} = 10$. **$p < 0.01$, ***$p < 0.001$, ****$p < 0.0001$, #$p$ vs. AAV::GFP during the same behavioral session (treatment level). On the scatterplot # indicates the axis-bound parameter.

The online version of this article includes the following source data and figure supplement(s) for figure 3:

*Figure 3 continued on next page*

*Figure 3 continued*

**Source data 1.** Related to *Figure 3B,D* and *Figure 3—figure supplement 2Bi,Ci,Di,Ei*.

**Figure supplement 1.** Histological evaluation of STN optogenetic viral injections.

**Figure supplement 2.** STN optogenetic manipulation of Go/No-Go (GNG) task parameters.

**Figure supplement 2—source data 1.** Related to *Figure 3—figure supplement 2Bii, Cii, Dii, Eii*.

## Pharmacological manipulation of mGlu4 interacts with trait impulsivity at STN

Altered glutamatergic neurotransmission is implicated in the pathobiology of impulsivity-related mental disorders (*Javitt, 2004*; *Jun et al., 2014*; *Sanacora et al., 2012*). Moreover, pharmacological modulation of both ionotropic and metabotropic glutamate receptors affects impulsivity (*Nikiforuk et al., 2010*; *Paine et al., 2007*; *Semenova and Markou, 2007*; *Sukhotina et al., 2008*). The metabotropic glutamate receptor 4 (mGlu4) emerged as a strong candidate, as it is widely expressed in the STN and the globus pallidus, which are both elements of the indirect pathway of movement in the basal ganglia (*Bradley et al., 1999*; *Corti et al., 2002*; *Iskhakova and Smith, 2016*; *Kristensen et al., 1993*; *Messenger et al., 2002*; *Testa et al., 1994*). Presynaptic activation of mGlu4 reduces, but does not abrogate, neurotransmitter release, making fast glutamatergic neurotransmission accessible to neuromodulatory therapeutic intervention. In fact, positive allosteric modulators (PAMs) of mGlu4 show promising results in pre-clinical and clinical trials as potential therapeutic agents to reverse motor dysfunction in Parkinson's disease (PD) (reviewed in *Charvin, 2018*; *Hopkins et al., 2009*), but can increase impulsivity in rats (*Isherwood, 2017*). We hypothesized that mGlu4 can specifically gate impulsive action via glutamatergic modulation of STN output and in consequence the expression of impulsive traits.

To investigate the possible role of mGlu4 in impulsivity, we used a PAM specific for this receptor, 4-((E)-styryl)-pyrimidin-2-ylamine (mGlu4 PAM; *East et al., 2010*; *Isherwood, 2017*). We delivered mGlu4 PAM to HI/LI animals and assayed for trait-dependent modulation of impulsivity (*Figure 4—figure supplement 1A*). This PAM increased the precue response rate (*Figure 4A–Ci*), without overtly affecting FA (*Figure 4A–Cii*, *Figure 4—figure supplement 1B*,C left) or Go responses (*Figure 4Ciii*, *Figure 4—figure supplement 1B*,C right) in the GNG task predominantly in LI animals. Thus, the effect appears to depend on baseline trait impulsivity.

To investigate the possible neuromodulatory interaction of mGlu4 with trait impulsivity in the STN circuitry, we treated HI and LI animals with mGlu4 PAM and then evaluated its effects throughout the brain by rs-fMRI. As for the comparison of HI and LI animals (*Figure 1*), we used a node-centric approach to rank-order hotspots of mGlu4 x HI/LI interaction. Again, rather than reporting p-value statistics, we reported the data as F values of the interactions and filtered for significance of the small sample size. Using this approach, we could directly map the interaction between mGlu4 modulation of brain functional connectivity and impulsive behavioral traits. As expected from the broad distribution of mGlu4 in the limbic system, mGlu4 PAM affected many brain areas (*Figure 4—figure supplement 2*), resulting from drug action on this distributed mGlu4 expression. Thus, this method should be sensitive to monitor the interaction of mGlu4 PAM treatment and HI/LI animals, which should reveal hotspots relevant for mGlu4 modulation of trait impulsivity. This node-wise analysis of functional connectivity ranked the STN as the brain node with the highest impulsivity x treatment interaction (*Figure 4D–Ei*, *Figure 4—figure supplement 2A-Di*). Rank analysis of this interaction, together with group splits for distance traveled (*Figure 4—figure supplement 1Dii*) and licks (*Figure 4—figure supplement 1Diii*), showed that the effect on the STN is specific for impulsivity (*Figure 4E*). Both the AMY and ZI ranked much lower in the interaction score for impulsivity than in the HI/LI functional connectivity split, indicating another role for mGlu4 in these structures.

## Pharmacological activation of mGlut4 modulates neuronal activity in the STN and SNr

To assess how the mGlu4 PAM network effects are reflected at the cellular level, we compared STN activity between vehicle and mGlu4 PAM treatment. The mGlu4 PAM reduced the fraction of units inhibited upon precue poke when compared to the controls (*Figure 5A and C*), and this fraction correlated negatively with waiting impulsivity (putative behavioral gating units; *Figure 2Bi*, D–E). We

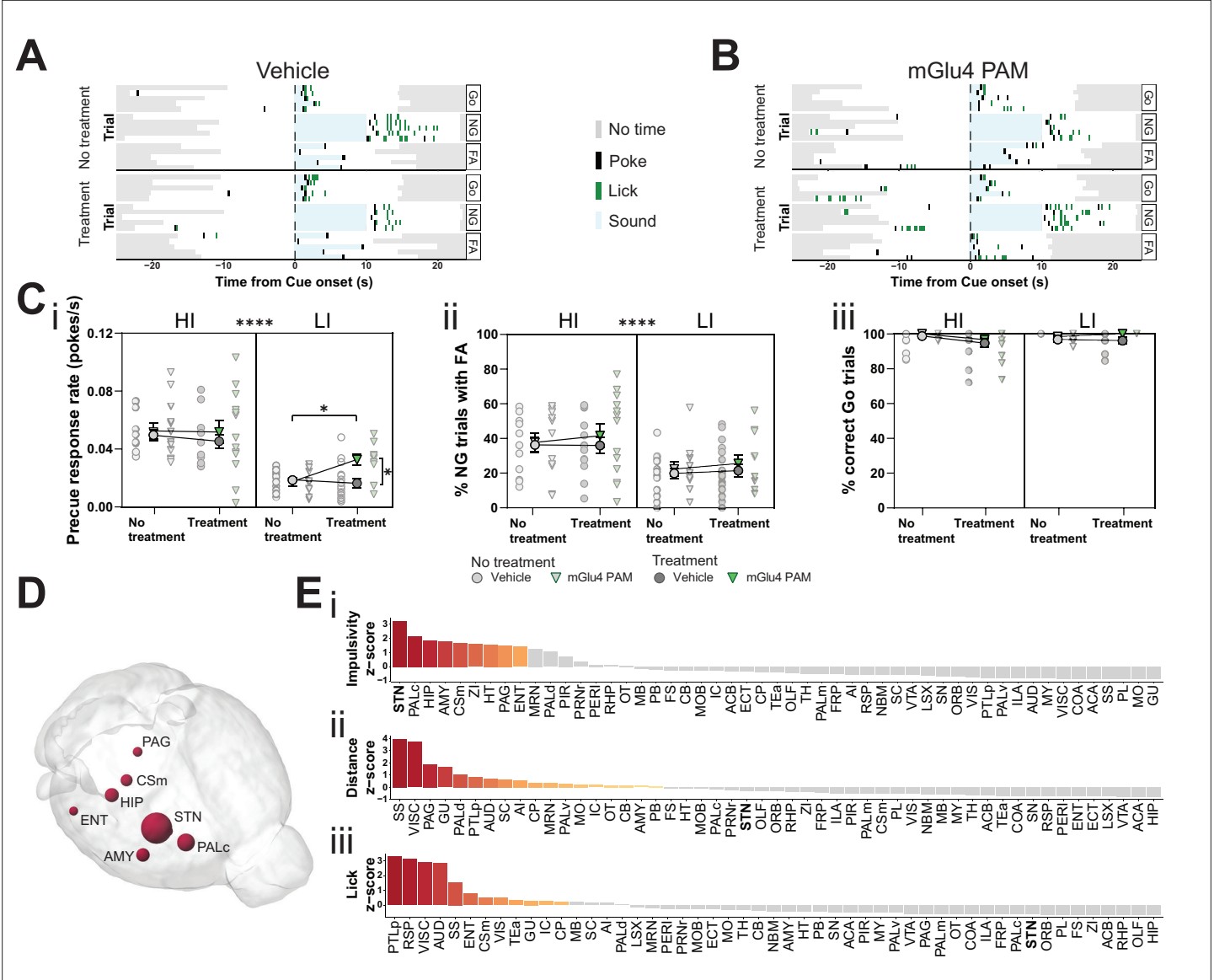

**Figure 4.** Metabotropic glutamate receptor 4 (mGlu4) positive allosteric modulator (PAM) modulates impulsivity and STN neuronal activity. (**A–B**) Example behavioral traces upon vehicle (**A**) or mGlu4 PAM (**B**) treatment in the correct Go (Go), correct No-Go (NG), and NG trials with false alarm (FA). For each example, top graph shows no treatment session, and bottom shows a behavior session with drug treatment for a single animal. Each row represents a single trial. (**C**) Systemic administration of mGlu4 PAM increases waiting impulsivity (**i**) in low impulsive (LI) animals (three-way repeated measure (RM) ANOVA with a main effect on impulsivity ($F_{1,50}$ = 62.72, p < 0.0001) and a phase × treatment interaction ($F_{1,50}$ = 4.217, p = 0.045)), without affecting the FA rate (three-way RM ANOVA with a main effect on impulsivity ($F_{1,50}$ = 14.34, p = 0.0004)) (**ii**) and % correct Go responses (three-way RM ANOVA with a phase × impulsivity interaction ($F_{1,49}$ = 6.507, p = 0.0139)) (**iii**). HI $N_{Vehicle}$ = 13 and $N_{mGlu4\ PAM}$ = 13, LI $N_{Vehicle}$ = 17 and $N_{mGlu4\ PAM}$ = 11. One animal each was removed in (**i**, HI/mGlu4 PAM) and (**iii**, LI/mGlu4 PAM) after Grubb's outlier test (alpha = 0.0001). (**D–E**) Brain-wide resting-state functional magnetic resonance imaging (rs-fMRI) screen for the interaction between mGlu4 PAM treatment and selected behavioral parameters in functional connectivity. (**D**) 3D visualization of the node-wise interaction score from a two-way RM ANOVA on treatment × impulsivity (size is correlated to F value) showing STN as the hotspot between the two. Only significantly scored nodes are shown. (**E**) Ordered, normalized two-way RM ANOVA F values from node-wise connectivity difference on interactions between treatment and group split by either impulsivity parameters (top), total distance traveled (middle), or number of licks (bottom) in the GNG task. Gray bars indicate that the p-value did not reach significance (Bonferroni corrected for multiple comparisons), whereas increased and significant differences are indicated in red. *p < 0.05, ****p < 0.0001.

The online version of this article includes the following source data and figure supplement(s) for figure 4:

**Source data 1.** Related to *Figure 4C*.

**Figure supplement 1.** Metabotropic glutamate receptor 4 (mGlu4) positive allosteric modulator (PAM) modulates trait impulsivity parameters.

**Figure supplement 1—source data 1.** Related to *Figure 4—figure supplement 1B, C*.

*Figure 4 continued on next page*

*Figure 4 continued*

**Figure supplement 2.** Brain-wide resting-state functional magnetic resonance imaging (rs-fMRI) screen for metabotropic glutamate receptor 4 (mGlu4) positive allosteric modulator (PAM) interaction.

**Figure supplement 2—source data 1.** Related to *Figure 4Ei* and *Figure 4—figure supplement 2B, C*.

observed no such effect in units gating the onset of immobility (*Figure 5B–C*). These data suggest that mGlu4 PAM specifically antagonizes STN activity to gate impulsive behavior.

A drop in LFP power in the STN correlates with precue pokes (*Figure 2G*). Thus, we reasoned that mGlu4 PAM should decrease LFP power in the STN. Consistent with the elevated precue response rateupon treatment with mGlu4 PAM (*Figure 4Ci*), LFP power in the beta and gamma band was reduced during the precue phase when compared to the vehicle control (*Figure 5Di*, inset). In the vehicle control, STN LFP power dropped prior to precue pokes (*Figure 5Dii* left), as observed above (baseline, *Figure 2Gi*), whereas after mGlu4 PAM treatment beta and gamma power were uncoupled from precue pokes (*Figure 5Dii* right). This mGlu4 PAM-mediated uncoupling was most evident when we compared the non-normalized absolute LFP power preceding precue pokes (left axes) in the vehicle control and after mGlu4 PAM treatment, and when analyzing the variance in raw LFP power around precue pokes (right axes, *Figure 5Diii*). This showed a significant reduction in STN LFP modulation during mGlu4 PAM treatment upon impulsive action. We propose that these changes reflect intra-STN modulation of STN neural activity and locally generated LFP (*Alavi et al., 2013*) by mGlu4.

The mGlu4 receptor is expressed mainly on pallido-subthalamo and subthalamo-nigral synapses (*Bradley et al., 1999*; *Corti et al., 2002*; *Iskhakova and Smith, 2016*; *Kristensen et al., 1993*; *Messenger et al., 2002*; *Testa et al., 1994*), making it well situated to gate neuronal processing through the STN (*Lanciego et al., 2012*), either by acting on GP inputs or at STN outputs to the substantia nigra pars reticulata (SNr). Furthermore, mGlu4 PAMs modulate subthalamo-nigral pathways ex vivo (*Valenti et al., 2005*; *Valenti et al., 2003*). Thus, we hypothesized that mGlu4 PAMs might modulate STN outputs to affect impulsivity in vivo. To test this hypothesis, we performed an electrophysiological characterization of SNr (*Figure 5—figure supplement 1*). Unlike our findings with the STN, we saw no responses in the SNr coupled to light onset (*Figure 5—figure supplement 2Ai*); however, we found event-coupled units to Go sound (*Figure 5—figure supplement 2Aii*), NG sound followed by FA (*Figure 5—figure supplement 2Aiii*), rewards (*Figure 5—figure supplement 2Aiv*), correct withhold (*Figure 5—figure supplement 2C*), and units bound to behavioral events such as to precue pokes (*Figure 5—figure supplement 2Bi*), Go pokes (*Figure 5—figure supplement 2Bii*), FA pokes (*Figure 5—figure supplement 2Biii*), immobility (*Figure 5—figure supplement 2Biv*), and movement onset (*Figure 5—figure supplement 2D*). In contrast to neurons in the STN, the firing rate of SNr neurons remained constant when trials were split by the number of precue responses (*Figure 5—figure supplement 2E*, see *Figure 2E* for comparison). Moreover, when compared to the population in the STN, the SNr population activity in PCA space showed less separated paths for behavior (*Figure 5—figure supplement 2F*) and for stimulus-centered events (*Figure 5—figure supplement 2H*).

As in the STN, SNr cell-by-cell analysis revealed differential encoding of precue pokes and Go sound responses, but to a lesser degree than in STN (*Figure 5—figure supplement 2G*), as for event onsets (*Figure 5—figure supplement 2I*). This difference between areas might reflect the functional proximity of the SNr to basal ganglia motor output, where impulsivity and motor commands may separate less than they do in the STN, which encodes traces of higher cognitive decisions. To investigate this possibility at the SNr network level, we inspected event- (precue period, Go, and NG sound periods) and behavior- (immobility) related spectral LFP in our task. SNr beta power increased during precue phases (*Figure 5—figure supplement 3A*) and reflects beta band synchrony between the STN and SNr, generally associated with behavioral inhibition (*Alavi et al., 2013*). Consistent with the proximity of the SNr to basal ganglia motor output, SNr gamma power transiently increased preceding execution of an action (*Figure 5—figure supplement 3B-C, Dii*). The increased beta power seen upon the NG cue reflects SNr behavioral inhibition recruited by the withhold signals in the task (*Figure 5—figure supplement 3Diii*). The strong motor binding of SNr oscillatory activity (as opposed to STN activity, *Figure 2Gi*) suggests SNr modulation by sources other than the STN (*Figure 5—figure supplement 3B*).

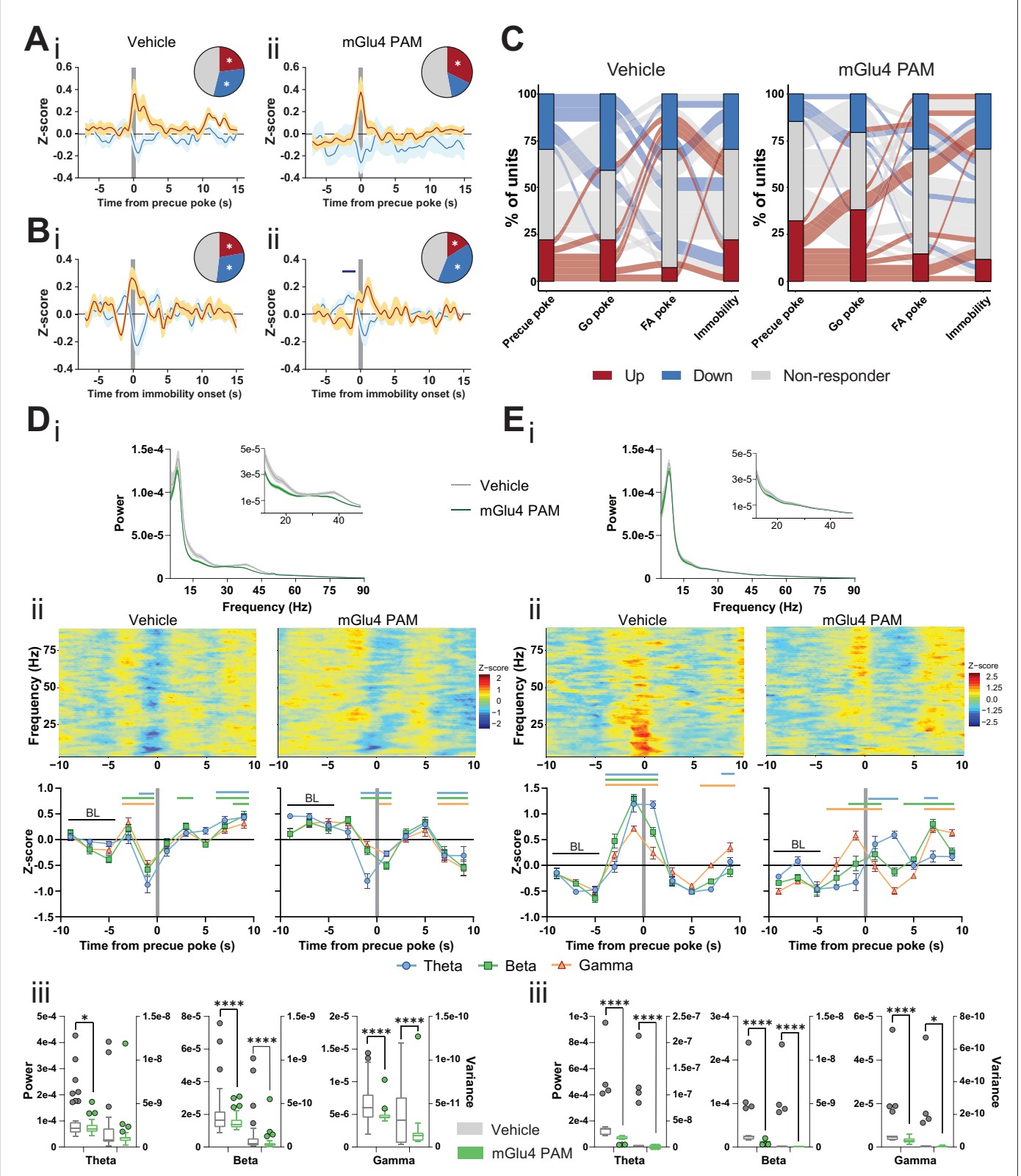

**Figure 5.** Metabotropic glutamate receptor 4 (mGlu4) positive allosteric modulators (PAM) modulates STN and substantia nigra pars reticulata (SNr) neuronal activity. (A–B) Peri-event data from in vivo extracellular recordings during the Go/No-Go (GNG) task showing unit responses aligned to onset of precue poke (A) or immobility (B) for vehicle (i) and mGlu4 PAM (ii) treatment. Population traces of excited and inhibited units for each line ($n_{\text{Precue, Vehicle, Up}} = 6$, $n_{\text{Precue, Vehicle, Down}} = 8$, $n_{\text{Precue, mGlu4 PAM, Up}} = 11$, $n_{\text{Precue, mGlu4 PAM, Down}} = 5$; $n_{\text{Immobility, vehicle, Up}} = 6$, $n_{\text{Immobility, Vehicle, Down}} = 8$, $n_{\text{Immobility, mGlu4 PAM, Up}} = 4$, $n_{\text{Immobility, mGlu4 PAM, Down}}$

*Figure 5 continued on next page*

*Figure 5 continued*

= 10, from N$_{Animals}$ = 3). The circles represent the proportion of cells excited (red), inhibited (blue) or non-responding to given event (* given population reached significance in permutation test). (**C**) Alluvial plot for individual STN units to precue, Go and false alarm (FA) pokes, and immobility onsets upon treatment. The width of the ribbon is proportional to the fraction of units with the given response pattern. Ribbons are color coded according to precue poke responses. (**D**) (**i**) LFP power spectra in STN during precue periods during vehicle or mGlu4 PAM treatment. (**ii**) Within-frequency Z-scored LFP power spectra (upper) and frequency band-averaged, 2s-binned (lower) time courses aligned to precue pokes during vehicle (left) or mGlu4 PAM treatment (right). Two-way repeated measures (RM) ANOVA revealed a significant main effect of time ($F_{7,966}$ = 43.23, p < 0.0001), no main effect of frequency band ($F_{2,138}$ = 1.449, p = 0.238), but a significant time × frequency band interaction ($F_{14,966}$ = 2.054, p = 0.012) for vehicle, and a significant main effect of time ($F_{7,966}$ = 30.65, p < 0.0001), no main effect of frequency band ($F_{2,138}$ = 1.724, p = 0.182), but a significant time × frequency band interaction ($F_{14,966}$ = 2.734, p = 0.0006) for mGlu4 PAM treatment. Mean Z-score ± SEM from n$_{Channels}$ = 48, N$_{Animals}$ = 3. Colored lines indicate significant differences to the baseline (BL) period in the respective frequency band, as determined by two-way RM ANOVA and Dunnet post hoc analysis (p < 0.05). (**iii**) LFP power of the time bin preceding precue pokes (left axis) and LFP power variance of the entire time range in (ii) (right axis) in the STN during vehicle and mGlu4 PAM treatment for theta, beta, and gamma frequency bands, respectively. Wilcoxon paired signed rank test (*p < 0.05, ****p < 0.0001). (**E**) (**i**) LFP power spectra in SNr during precue periods during vehicle or mGlu4 PAM treatment. (**ii**) Within-frequency Z-scored LFP power spectra (upper) and frequency band-averaged, 2s-binned (lower) time courses aligned to precue pokes during vehicle (left) or mGlu4 PAM treatment (right). Two-way RM ANOVA revealed a significant main effect of time ($F_{7,651}$ = 126.2, p < 0.0001), no main effect of frequency band ($F_{2,93}$ = 1.395, p = 0.253), but a significant time × frequency band interaction ($F_{14,651}$ = 10.34, p < 0.0001) for vehicle, and significant main effects of time ($F_{7,651}$ = 25.87, p < 0.0001) and frequency band $F_{2,93}$ = 3.772, p = 0.027), and a significant time × frequency band interaction ($F_{14,651}$ = 11.99, p < 0.0001) for mGlu4 PAM treatment. Mean Z-score ± SEM from n$_{Channels}$ = 32, N$_{Animals}$ = 2. Colored lines indicate significant differences to the baseline (BL) period in the respective frequency band, as determined by two-way RM ANOVA and Dunnet post hoc analysis (p < 0.05). (**iii**) LFP power of the time bin preceding precue pokes (left axis) and LFP power variance of the entire time range in (ii) (right axis) during vehicle and mGlu4 PAM treatment in the SNr for theta, beta, and gamma frequency bands, respectively. Wilcoxon paired signed rank test (*p < 0.05, ****p < 0.0001).

The online version of this article includes the following source data and figure supplement(s) for figure 5:

**Source data 1.** Related to *Figure 5A, B, D and E*.

**Figure supplement 1.** Electrode placement.

**Figure supplement 2.** Encoding Go/No-Go (GNG) task parameters in substantia nigra pars reticulata (SNr).

**Figure supplement 2—source data 1.** Related to *Figure 5—figure supplement 2A–E*.

**Figure supplement 3.** LFP time courses for Go/No-Go (GNG) task parameters in substantia nigra pars reticulata (SNr).

**Figure supplement 3—source data 1.** Related to *Figure 5—figure supplement 3A–D*.

**Figure supplement 4.** Metabotropic glutamate receptor 4 (mGlu4) positive allosteric modulator (PAM) interacts with substantia nigra pars reticulata (SNr) activity in the Go/No-Go (GNG) task.

**Figure supplement 4—source data 1.** Related to *Figure 5—figure supplement 4A, B*.

Since the mGlu4 receptor is known to modulate synaptic transmission between the STN and SNr, we reasoned that the effects of mGlu4 PAM in the SNr should be particularly pronounced in the SNr. Indeed, mGlu4 PAM treatment modulated SNr recruitment to precue pokes (*Figure 5—figure supplement 4A*,C), but not its binding to basic motor output (*Figure 5—figure supplement 4B*,C). At the level of LFPs, we observed only a minor reduction in overall LFP power upon mGlu4 PAM treatment (*Figure 5Ei*). However, this was paralleled with a significant uncoupling of theta and beta band coupling to precue pokes in the SNr (*Figure 5Eii*), and in the variance of raw LFP power around precue pokes (*Figure 5Eiii*).

The LFP power spectra and their modulation by mGlu4 PAM (*Figure 5Di*, Ei) explain the dominant interaction of STN with impulsive choice, compared to SNr (*Figure 1Ei*) and its interaction with mGlu4 modulation (*Figure 4Ei*). Together, these data suggest that mGlu4 modulates the control of impulsive action by the STN and SNr, without altering activity bound to motor output.

## STN mGlu4 dissociates impulsive traits from motor function

We found that the STN is a hotspot for the interaction of mGlu4 PAM activity with trait impulsivity, strongly suggesting that mGlu4 expression in the STN modulates this behavioral trait directly. We examined animals with the HI or LI phenotype (similar group separation to our rs-fMRI experiment) in our GNG task (*Figure 6—figure supplement 1A*) and performed high-resolution in situ hybridization to locate areas of mGlu4 expression in samples from both groups (*Figure 6A*). Quantification of the in situ hybridization signal revealed a statistically significant difference in *Grm4* mRNA expression in the STNs of HI and LI animals, but not in the other brain regions we analyzed, including the ZI, thalamus, and AMY (*Figure 6B*). As expected, the predominant signal in the perinuclear portion

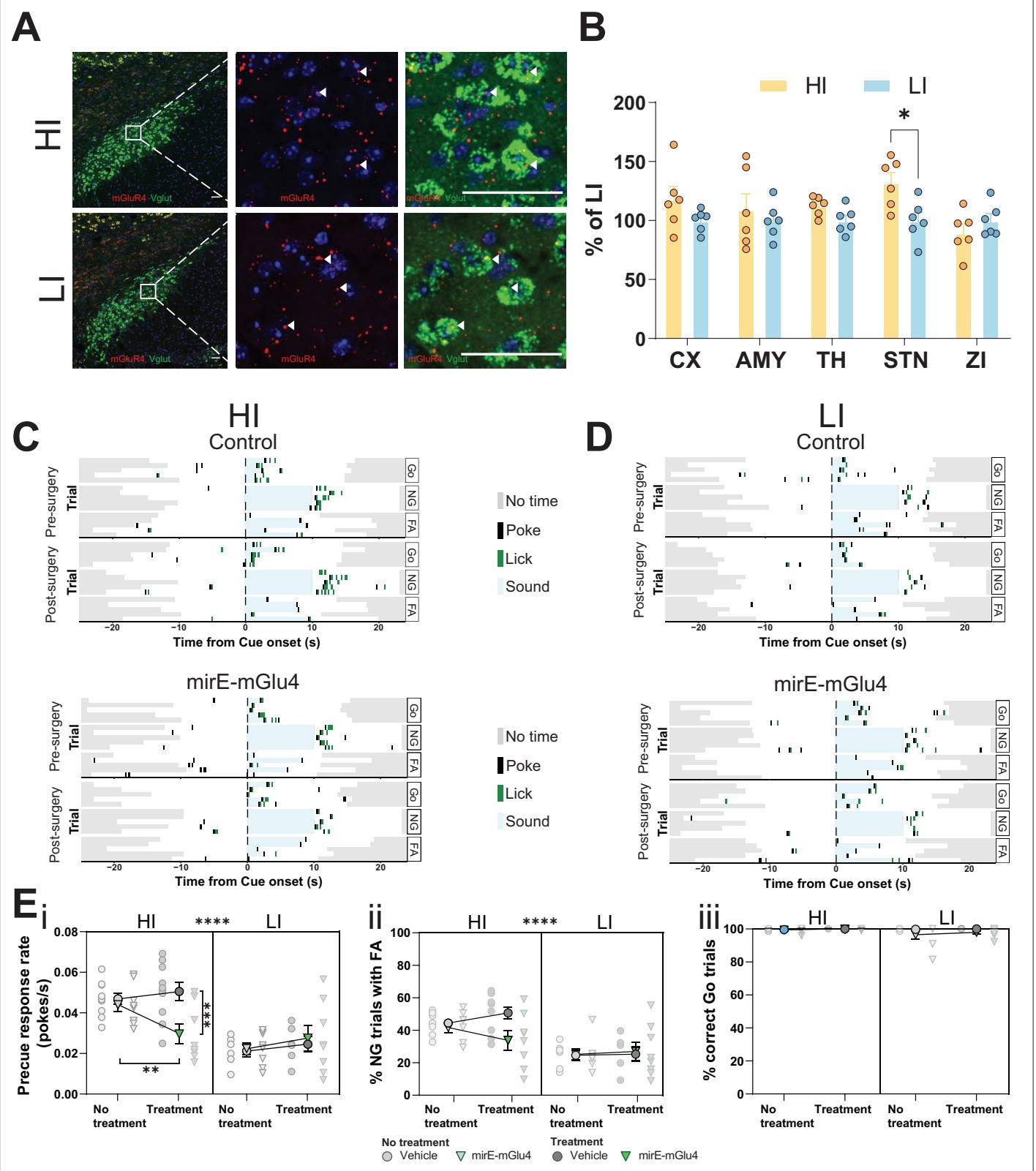

**Figure 6.** Subthalamic nucleus (STN) metabotropic glutamate receptor 4 (mGlu4) dissociates impulsive traits from motor function. (**A**) STN brain sections from high impulsive (HI) and low impulsive (LI) animals analyzed by in situ mRNA hybridization using probes against vesicular glutamate transporter (VGlut) (*Slc17a6/Slc17a7*, green) and mGlu4 (red) transcripts. White arrows mark positive cells for receptor expression. Scale bars = 40 μm. (**B**) Two-way repeated measures (RM) ANOVA of the signal quantification between somatosensory cortex (CX), basolateral nucleus of amygdala (AMY),

*Figure 6 continued on next page*

*Figure 6 continued*

ventral anterior-lateral complex of the thalamus (TH), STN, and zona incerta (ZI) revealed no significant main effect of impulsivity level ($F_{1,40}$ = 2.927, p = 0.118), but a significant main effect of brain area ($F_{4,40}$ = 2.841, p = 0.037), and interaction between the two ($F_{4,40}$ = 2.841, p = 0.037), with a significant increase of mGlu4 receptor only in the STN of HI animals compared to their LI littermates ($n_{sections}$ = 6, $N_{animals}$ = 3 per group). (**C–D**) Exemplary behavioral traces of single HI (**C**) and LI (**D**) animals, part of either the control group (top panels) or mirE-mGlu4 group (lower panels), in correct Go (Go), correct No-Go (NG), and NG trials with false alarm (FA). Top shows a session prior to surgery. Bottom shows a post-surgery behavior session. Each row represents a single trial. (**E**) mirE-mediated knockdown of mGlu4 decreases waiting impulsivity with a phase × impulsivity × virus interaction (three-way RM ANOVA, $F_{1,28}$ = 4.375, p = 0.046) (**i**), without significantly affecting FA rate (three-way RM ANOVA, main impulsivity effect ($F_{1,28}$ = 24.18, p < 0.0001)) (ii) or correct Go responses (three-way RM ANOVA, no main impulsivity effect ($F_{1,28}$ = 2.471, p = 0.127)) (iii). HI $N_{Control}$ = 10 and $N_{mirE-mGlu4}$ = 8, LI $N_{Control}$ = 6 and $N_{mirE-mGlu4}$ = 8. No animals were removed after Grubb's outlier test (alpha = 0.0001).

The online version of this article includes the following source data and figure supplement(s) for figure 6:

**Source data 1.** Related to *Figure 6B and E*.

**Figure supplement 1.** Knockdown of STN metabotropic glutamate receptor 4 (mGlu4) modulates impulsivity.

**Figure supplement 1—source data 1.** Related to *Figure 6—figure supplement 1A–C*.

**Figure supplement 2.** Histological evaluation.

**Figure supplement 3.** Knockdown of STN metabotropic glutamate receptor 4 (mGlu4) modulates impulsivity.

**Figure supplement 3—source data 1.** Related to *Figure 6—figure supplement 3B, C*.

**Figure supplement 4.** STN gates impulsive traits independent from general motor function.

**Figure supplement 4—source data 1.** Related to *Figure 6—figure supplement 3A–H*.

**Figure supplement 5.** Subject history of the different mouse cohorts used in this study.

**Figure supplement 6.** STN circuitry multiplexes motor function and impulsive traits as dissociable states.

**Figure supplement 7.** Distribution of precue response rates across behavioral cohorts.

**Figure supplement 7—source data 1.** Related to *Figure 6—figure supplement 7*.

(*Figure 6—figure supplement 1Bi*) and in VGlut$^+$ neurons (detected by a probe mix for *Slc17a6* and *Slc17a7*; *Figure 6—figure supplement 1Bii*) covaried most strongly with the HI and LI phenotypes. This suggests that locally synthesized mGlu4 is closely linked to trait impulsivity (as opposed to STN afferents also expressing mGlu4).

To study the function of mGlu4 in trait impulsivity, we used adeno-associated viral vectors expressing short hairpin (sh)RNAs against the receptor (mirE-mGlu4) (*Fellmann et al., 2013*; *Groessl et al., 2018*). First, we tested the efficacy of the vectors in Neuro-2a cell culture, a cell type known to express mGlu4 endogenously (*Hruz et al., 2008*). The shRNAs effectively lowered mGlu4 protein levels, as assayed by Western blot (*Figure 6—figure supplement 1C*). We injected the validated vectors into the STN (*Figure 6—figure supplement 2*) of HI and LI animals (*Figure 6—figure supplement 3A*). HI animals injected with mirE-mGlu4 showed a persistent decrease in precue response rate when compared with the control group post-surgery (*Figure 6Ei*, *Figure 6—figure supplement 3Bi*); this effect was absent in LI animals (*Figure 6Ei*, *Figure 6—figure supplement 3Ci*). As with pharmacological inhibition, we saw no changes in FA (*Figure 6Eii*) or Go responses (*Figure 6Eiii*, *Figure 6—figure supplement 3Bii,Cii*). Taken together, the data from our optogenetic, pharmacological, mGlu4 expression, and silencing experiments show complementary phenotypes in re-occurring patterns in HI and LI animals. These are consistent with mGlu4 in the STN modulating (waiting) impulsivity traits.

## mGlu4 manipulation in the STN does not affect general motor function

To compare the consequences of modulation of STN activity on impulsive behavior with those on general motor function (*Andrén et al., 1995*; *Eagle et al., 2008b*; *Karachi et al., 2009*; *Yasoshima et al., 2005*), we performed a series of open-field experiments using various means to manipulate STN function, as above. As expected, high-power optogenetic activation of the STN suppressed motor output (*Figure 6—figure supplement 4A*). However, low-power optogenetic activation of the STN, as used in the GNG task and akin to pharmacologically blocking GABA transmission to the STN (*Périer et al., 2002*), did not alter motor behavior. Optogenetic inhibition, which might be expected to be less effective than complete STN lesions (*Andrén et al., 1995*), modulated impulsivity in the GNG task independent of general motor effects (*Figure 6—figure supplement 4A*). Furthermore, both pharmacological (*Figure 6—figure supplement 4B*) and shRNA-mediated inhibition of mGlu4

activity (*Figure 6—figure supplement 4C*) affected impulsivity without affecting general motor behavior. Moreover, in a separate, untreated cohort of animals, we found no significant correlation between distance traveled in OF and either precue response rate (*Figure 6—figure supplement 4D*) or percent of trials with FA (*Figure 6—figure supplement 4E*). This finding is consistent with reports that general motor activity does not clearly correlate with impulsivity in different mouse strains (*Gubner et al., 2010*; *Molander et al., 2011*). Thus, we conclude that weak manipulations that specifically target impulsivity but not motor functions lead to the observed phenotype in the GNG task. We speculate that this differential behavioral effect reflects the separation of motor and impulsivity-related stimuli and behaviors in different regimes of STN activity (*Figure 2C*, *Figure 2—figure supplement 2G*, *Figure 6—figure supplement 6*).

## Discussion

Our study exploits the inherent variance in animal behavior to deconstruct neuronal circuit mechanisms and genetic factors that underlie trait impulsivity. By combining small animal fMRI with circuit neuroscience methods (*Kargl et al., 2020*; *Wank et al., 2021*), we were able to identify brain hotspots accounting for natural variance in trait impulsivity in an unbiased way. The STN emerges as the top ranked candidate, with higher functional connectivity in LI animals reflecting behavioral gating. Conversely, the lower level of STN functional connectivity in HI, we suggest, is a sign of less effective interactions or rapid transfer of information between and across the STN and the rest of the brain network. We validated our rs-fMRI screen with subsequent electrophysiological profiling and with optogenetic, pharmacological, and genetic manipulations. These mechanistic analyses reveal that the STN circuitry dissociates basic motor functions from impulsive responses. Additionally, we find that both STN activity and mGlu4 expression depend on the intrinsic trait impulsivity of the subject. Our findings suggest that STN, a known node in basal ganglia motor processing, is also crucial for the control of trait impulsivity via mGlu4.

The identification of STN controlling trait impulsivity adds a novel dimension to previous studies on impulsivity and STN function. STN activity was previously shown to respond to stop cue presentations, contributing to behavioral inhibition (*Schmidt et al., 2013*). Additionally, it was functionally linked to Go and NG trial discrimination, and contributes to reactive control (*Pasquereau and Turner, 2017*) and behavioral choice (*Mazzoni et al., 2018*). Moreover, STN lesions and pharmacological studies link its function to impulsive action (*Baunez et al., 2001*; *Baunez and Robbins, 1997*; *Eagle et al., 2008b*; *Karachi et al., 2009*; *Uslaner and Robinson, 2006*; *Wiener et al., 2008*), whereas human fMRI and DBS studies confirmed its key role in motor control and implied role in ICDs (*Georgiev et al., 2016*; *Le Jeune et al., 2010*; *Pote et al., 2016*; *Pötter-Nerger et al., 2017*; *Yoon et al., 2019*). Here, we extended these findings on the role of the STN by identifying it as one of the main hotspots for trait impulsivity in a brain-wide screen. Consistent with this role, human fMRI and electrophysiology studies showed that the magnitude of STN activation during stop trials correlated inversely with both trait impulsivity and reaction time in GNG tasks (*Aron and Poldrack, 2006*; *Yoon et al., 2019*). We note, however, that these studies of human subjects were performed mainly on patients with PD (*Le Jeune et al., 2010*; *Mazzoni et al., 2018*; *Voon et al., 2017*; *Zavala et al., 2018*; *Zavala et al., 2017*). Here, we demonstrate a specific role of the STN in controlling trait impulsivity and find that impulsive action is an STN neuronal state that is separable from motor activity.

Our data reveal that the STN is engaged in one of the two main impulsivity parameters, the precue response rate over NG cue withholding (*Figure 2Aiii*,Bi,iii,C,D) in a free-moving, cue-initiated GNG task. We and others showed that the precue response rate and FAs are highly correlated, as both subserve behavioral inhibition and serve as reliable measures of impulsivity in GNG tasks (*Gubner et al., 2010*). Our data showed a significant correlation between precue response rate and the percentage of trials with FA (*Figure 6—figure supplement 4F*), but no correlation between the percentage of correct Go responses and either parameter (*Figure 6—figure supplement 4G*,H). This suggests independent processing of each task response. We argue that the precue responses may reflect anticipatory, premature behavioral execution (in the domain of waiting impulsivity); this responding before the reward is actually due is similar to the premature responding seen in five-choice serial-reaction time tasks (5-CSRT) (*Bari et al., 2008*; *Dalley and Robbins, 2017*; *Sanchez-Roige et al., 2012*). Premature responses in 5-CSRT and 4-CSRT tasks were previously interpreted as waiting impulsivity, a facet of impulsivity distinct from motor impulsivity (*Dalley and Ersche, 2019*; *Robinson*

*et al., 2009*), in both human and rodent studies (*Voon, 2014*). Moreover, this aspect of impulsivity was associated with the development of drug addiction, as well as an expression of attention deficit in ADHD (*Diergaarde et al., 2009*; *Morris et al., 2016*; *Robinson et al., 2009*). FAs indicate learning the instrumental omission contingency and thus are associated with inhibition of impulsive action (stopping impulsivity) (*Gubner et al., 2010*; *Moschak et al., 2013*; *Moschak et al., 2012*; *Moschak and Mitchell, 2012*). Based on our results, we conclude that the STN encodes the gating of this behavior through a subset of neurons inversely correlated to precue responses (*Figure 2D and E*). This was further confirmed by our optogenetic inhibition studies, which showed a significant change in the precue responses when STN activity was manipulated, with no effect on cue-related impulsivity (*Figure 3B and D*). Finally, our findings are consistent with previous human studies targeting the STN in relation to waiting impulsivity and the development of addictive behavior (*Aleksandrova et al., 2013*; *Morris et al., 2016*).

Previous pharmacological studies using systemic administration of an mGlu4 PAM found a link between impulsivity and mGlu4 activity (*Isherwood, 2017*). Here, we confirm and extend this finding by linking the action of the mGlu4 PAM to baseline impulsivity levels, showing a state-dependent effect. Moreover, we provide insights into the neuroanatomical substrates and

**Table 1.** Disease association score.
The disease-gene association scores for *Grm4*, filtered for psychiatric disorders that share the impulsivity domain. Data extracted from Open targets database (*Carvalho-Silva et al., 2019*) and DISEASES (*Pletscher-Frankild et al., 2015*). N.D. – not determined.

| Disorder | Open targets (Overall association score) | Diseases (Z-score) |
|---|---|---|
| *Parkinson's disease* | 0.070 | 4.7 |
| *Schizophrenia* | 0.061 | 4 |
| *Mood disorder* | 0.064 | 3.1 |
| *Anxiety disorder* | 0.066 | 2.4 |
| *Autistic disorder* | 0.015 | 1 |
| *Schizoaffective disorder* | 0.015 | 1.5 |
| *Movement disease* | 0.058 | 1.5 |
| *Drug/alcohol dependence* | 0.016 | 1.4 |
| *Alzheimer's disease* | N.D. | 1.3 |
| *Withdrawal disorder* | N.D. | 1.3 |
| *Unipolar depression* | 0.063 | N.D. |
| *Nervousness* | 0.040 | N.D. |
| *Bipolar disorder* | 0.019 | N.D. |

mechanism of action underlying the drug effect. Despite the widespread expression of mGlu4 in the brain, we demonstrate by rs-fMRI analysis that the STN is the crucial node in the interaction between mGlu4 PAM and trait impulsivity. In this context, our electrophysiological and rs-fMRI data sample mGlu4 modulation upstream and downstream of the STN. To assess postsynaptic effects within those networks, we analyzed the activity of one of the main STN outputs, the SNr. These recordings revealed encoding of several GNG features within that nucleus (*Figure 5—figure supplement 2A-B*), along with strong binding to immobility and movement onset (*Figure 5—figure supplement 2Biv,D*), consistent with SNr motor functions. Since dissociating impulsivity and motor features deteriorates from the STN to the SNr, this result may indicate their proximity to cognitive function and basal ganglia output, respectively (*Aron et al., 2016*; *Bryden et al., 2016*; *Deniau et al., 2007*; *Jantz et al., 2017*; *Rektor et al., 2015*; *Weintraub and Zaghloul, 2013*; *Zavala et al., 2018*). These findings are mirrored by rs-fMRI results that are less distinct between HI and LI in SNr compared to STN (*Figure 1D and E*). Notably, mGlu4 directly attenuates temporal dynamics (variance) in STN/SNr LFP power. We interpret this as a signature of uncoupling of the STN/SNr from interconnected functional mesoscale networks, so affecting behavioral inhibition during precue phases. This puts into context studies that link STN/SNr connectivity to motor behavior in humans, with a pronounced bias toward waiting impulsivity (*Alavi et al., 2013*). That the mGlu4 PAM was efficacious following systemic administration further strengthens the idea that mGlu4 might serve as a therapeutic target for the treatment of maladaptive impulsivity in psychiatric patients. Together, we provide mechanistic insights into the dynamics of how mGlu4 PAMs modulate brain circuitry at the level of STN.

The gene encoding human mGlu4, *Grm4*, has not previously been implicated in impulsivity phenotypes, neither by findings of single nucleotide polymorphisms associated with impulsive disorders nor by differences in expression levels in humans. Meta-analyses and genome-wide association studies

(GWAS), however, have found associations between *Grm4* and several psychiatric disorders that share the impulsivity domain (see *Table 1*), such as PD, bipolar disorder, and ADHD (*Carvalho-Silva et al., 2019*; *Fliers et al., 2012*; *Pletscher-Frankild et al., 2015*). Some of these studies link to SNPs at miRNA binding sites at mGlu4 or miR (micro-RNA) of which mGlu4 is target of, while both could affect gene expression. These data, together with human fMRI studies, suggest that mGlu4 modulation of STN function may contribute to impulsivity in humans. Here, we demonstrate that silencing of *Grm4* specifically in the STN decreases the impulsive behavior of HI animals (*Figure 6C and E*; *Figure 6— figure supplement 3B*). While we cannot exclude the contributions of upstream mGlu4 PAM target regions, our results indicate a presynaptic site of action downstream of STN driving the impulsivity phenotype. In this regard, we observed significantly lower mGlu4 levels in rodents selected for low trait impulsivity compared to HI rodents (*Figure 6B*). As mGlu4 modulation of STN efferent activity has been demonstrated in vitro, we speculate that high mGlu4 expression decreases STN signaling and fine-tunes the inhibition of impulsivity by the STN.

In summary, by combining rs-fMRI, electrophysiology, optogenetics, pharmacology, and genetics, we have discovered a previously unknown role of mGlu4 function in the STN as a crucial modulator of trait impulsivity. Synaptic modulation by metabotropic glutamate in the STN dissociates and fine-tunes impulsive traits independent of general motor function. We propose that mGlu4-driven neuromodulation of STN activity regulates impulsive states and biases toward impulsive action, while remaining subthreshold for affecting representation and execution of gross motor gating (*Figure 2Cii*). In this way, the brain independently and selectively modulates a cognitive behavioral trait within a basic motor circuit by using metabotropic glutamate signaling (*Figure 6—figure supplement 6*). This demonstrates that multiplexing of different functions (here, impulse control and general motor gating) within the same network via neuromodulation (*Bargmann, 2012*) allows the brain to control and adapt multiple behaviors with limited circuitry. From a translational perspective, the STN, mGlu4, and metabotropic glutamate may emerge as potential targets for impulsivity treatments. This opens opportunities for therapeutic interventions that selectively target pathological impulsivity without affecting general motor performance.

## Limitations of the study

Trait differences within an isogenic strain allow us to discover environmental factors and/or epigenetic changes that play a role in behavior. We focused in this study on mGlu4 because human GWAS indirectly linked the *GRM4* gene with impulsivity and provided a rationale for our investigations (*Table 1*). Exploring inter-individual variation in impulsive behavior in an isogenic animal model, however, cannot address the entire spectrum of genomic variations that might influence impulsive traits, particularly in the human context.

We recognize the limitations of the study due to the use of male subjects only. This is not unique to this impulsivity study (*Bellés et al., 2021*; *Jung et al., 2020*; *Zapata and Lupica, 2021*). We chose to study males not only because surveys by the World Health Organization show a significant gender bias toward males in the prevalence of ADHD (*Fayyad et al., 2017*) but also to allow us to compare our findings to those of previous studies cited in this paper (*Isherwood, 2017*). Moreover, we expected that genetic homogeneity in our test subjects would improve the statistical power of the study, given the constraints in sample sizes. That said, our study paves the way for important future studies to explore if and how the mechanism identified here differs in females.

Finally, additional data on the effects of mGlu4 PAM delivered directly to the brain might further support the mechanism we propose. At this point, however, the properties of the compound do not allow direct brain parenchymal infusions, for that purpose.

# Materials and methods

## Key resources table

| Reagent type (species) or resource | Designation | Source or reference | Identifiers | Additional information |
|---|---|---|---|---|
| Strain, strain background (*Mus musculus*, male) | Wild-type | Jackson Laboratory | | C57BL/6J background |

*Continued on next page*

*Continued*

| Reagent type (species) or resource | Designation | Source or reference | Identifiers | Additional information |
|---|---|---|---|---|
| Cell line (*Mus musculus*) | Neuro-2a | ATCC | CCL-131 | |
| Other | mGlu4-shRNA (mirE-mGlu4) | This paper | AAV2/5.SFFV.GFP.mGlu4-miR-E.WPRE | AAV vectors to transduce brain tissue; Titer: $4.24 \times 10^{13}$ |
| Other | Renilla-Control shRNA (mirE-Renilla) | This paper | AAV2/5.SFFV.GFP.Renilla-miR-E.WPRE | AAV vectors to transduce brain tissue; Titer: $1.86 \times 10^{13}$ |
| Other | syn-GFP | Penn Vector Core | AAV5.hsyn.eGFP.WPRE.hGH | AAV vectors to transduce brain tissue; Titer: $1.15 \times 10^{13}$ |
| Other | syn-ChR2 | Penn Vector Core/ Addgene | AAV2/5.hsyn.hChR2(H134R).eYFP.WPRE | AAV vectors to transduce brain tissue; Titer: $1.3 \times 10^{13}$ |
| Other | syn-Arch | Penn Vector Core | AAV5.hsyn.ArchT.YFP.WPRE.hGH | AAV vectors to transduce brain tissue; Titer: $4.68 \times 10^{12}$ |
| Chemical compound, drug | DAPI | Life Technologies | DAPI | 1 µg/ml |
| Chemical compound, drug | mGlu4 PAM | Boehringer Ingelheim | 4-((*E*)-styryl)-pyrimidin-2-ylamine | 80 mg/kg |
| Commercial assay or kit | RNAscope Multiplex Fluorescent v2 kit | Advanced Cell Diagnostics | Cat no. 323100 | |
| Commercial assay or kit | Proprietary probes against *Gmr4* (mGlu4) | Advanced Cell Diagnostics | Cat no. 480991 | |
| Commercial assay or kit | Proprietary probes against *Slc17a6+ Slc17* a7 (VGlut2 +1) | Advanced Cell Diagnostics | Cat no. 416631 and 319171 | |
| Commercial assay or kit | TMR Fluorescein Evaluation kit | Perkin Elmer | Cat no. NEL 760001KT | |
| Software, algorithm | GraphPad Prism 7&8 | GraphPad Software, Inc | Version 8.1.1 | |
| Software, algorithm | scikit-learn package | doi:10.1007/s13398-014-0173-7.2 | Python 3 | |
| Software, algorithm | MATLAB | Mathworks | R2015b | |
| Software, algorithm | Ethovision | Noldus Information Technology | XT 8 and 12 | |
| Software, algorithm | AnyMaze | Stoelting | | |
| Software, algorithm | TSE VideoMot 3D | TSE Systems | Version 7.01 | |
| Software, algorithm | Offline Sorter | Plexon | Version 4 | |
| Software, algorithm | CinePlex Studio & Editor | Plexon | Version 3.6 | |
| Software, algorithm | Neuroexplorer | Plexon | Version 5 | |
| Software, algorithm | Clampfit software | Molecular Devices | | |
| Software, algorithm | Omniplex | Plexon | Version 1.16.2 | |
| Software, algorithm | R | The R Project | Version 3.4 | |
| Software, algorithm | DPABI | DPABI R-fMRI Network | Version 2.1 | |
| Software, algorithm | Paravision | Bruker | Version 6.1 | |
| Software, algorithm | Adobe Illustrator | Adobe | RRID:SCR_010279 | |
| Sequence-based reagent | Grm4.1332 | EntrezID: 268,934 Guide: TTCTGATG TACTTAAGCAGCTG | Gene-specific sequences for mGlu4-shRNA knock down | 97mer: TGCTGTTGACAGTGAGCGAAGCT GCTTAAGTACATCAGAATAGTGAAGCCAC AGATGTATTCTGATGTACTTAAGCAGCTG TGCCTACTGCCTCGGA |

*Continued on next page*

*Continued*

| Reagent type (species) or resource | Designation | Source or reference | Identifiers | Additional information |
|---|---|---|---|---|
| Sequence-based reagent | Grm4.337 | EntrezID: 268,934 Guide: TTAGAGACCCAT GAATAGCGGG | Gene-specific sequences for mGlu4-shRNA knock down | 97mer: TGCTGTTGACAGTGAGCGACCGCTAT TCATGGGTCTCTAATAGTGAAGCCACAGATG TATTAGAGACCCATGAATAGCGGGTGCCTA CTGCCTCGGA |
| Sequence-based reagent | Grm4.2087 | EntrezID: 268,934 Guide: TTGACAAT GGGTATGGGCTGGC | Gene-specific sequences for mGlu4-shRNA knock down | 97mer: TGCTGTTGACAGTGAGCGACCAGCC CATACCCATTGTCAATAGTGAAGCCACAGAT GTATTGACAATGGGTATGGGCTGGCTGCCT ACTGCCTCGGA |
| Sequence-based reagent | Grm4.788 | EntrezID: 268,934 Guide: TTTGATGA TCTTGTCAAACTCC | Gene-specific sequences for mGlu4-shRNA knock down | 97mer: TGCTGTTGACAGTGAGCGAGAGTTTGA CAAGATCATCAAATAGTGAAGCCACAGA TGTATTTGATGATCTTGTCAAAC TCCTGCCTACTGCCTCGGA |

## Subjects

C57Bl6/J male mice purchased from Jackson Laboratory were used in all experiments. Animals were housed in groups of max 5 and kept under a 12:12 hr light:dark cycle with food and water ad libitum. Experiments were conducted during the light period. Water deprivation was practiced over the period of GNG behavioral testing with exception of the post-surgery recovery period. All animal experiments were performed in accordance with institutional guidelines and were approved by the respective Austrian (BGBl nr 501/1988, idF BGBl I no 162/2005) and European (Directive 86/609/EEC of November 24, 1986, European Community) authorities and covered by the license GZ2452882016/6. Overall animal history is shown in *Figure 6—figure supplement 5*.

## Ex vivo electrophysiology

Male wild-type mice (2–3 months of age) were deeply anesthetized with isofluorane, decapitated, and their brains quickly chilled in sucrose-based dissection buffer, bubbled with 95% $O_2$/5% $CO_2$ containing the following (in mM): 220 sucrose, 26 $NaHCO_3$, 2.4 KCl, 10 $MgSO_4$, 0.5 $CaCl_2$, 3 sodium pyruvate, 5 sodium ascorbate, and 10 glucose. Transverse coronal brain slices (300 µm) were cut in dissection buffer using a Vibratome (Leica, VT1000S) and immediately incubated for 15 min recovery phase in oxygenated artificial cerebrospinal fluid (aCSF) 126 NaCl, 2.5 KCl, 1.25 $NaH_2PO_4$, 26 $NaHCO_3$, 2.5 $CaCl_2$, 2.5 $MgCl_2$, and 25 glucose in 95% $O_2$/5% $CO_2$ at 32°C. This was followed by a slice resting phase with oxygenated aCSF for at least 45 min at room temperature (RT).

Individual brain slices containing STN were placed on the stage of an upright, infrared-differential interference contrast microscope (Olympus BX50WI) mounted on a X-Y table (Olympus) and visualized with a 40× water immersion objective by an infrared sensitive digital camera (Hamamatsu, ORCA-03). Slices were fully submerged and continuously perfused at a rate of 1–2 ml/min with oxygenated aCSF. Neurons, either infected by AAV::Arch or AAV::ChR2, were identified by the presence of YFP fluorescence. Patch pipettes were pulled on a Flaming/Brown micropipette puller (Sutter, P-97) from borosilicate glass (1.5 mm outer and 0.86 mm inner diameter, Sutter) to final resistances ranging from 3 to 5 MΩ. Internal solution for voltage-clamp recordings of responses to optogenetic stimulation contained (in mM): potassium gluconate, 135; KCl, 5; HEPES, 10; $MgCl_2$, 2; EGTA, 0.2; MgATP, 4; $Na_3GTP$, 0.4; $K_3$-phosphocreatine, 10; biocytin, 0.1; pH 7.2 (with KOH). Cells were held at –70 mV. Cells were allowed to reestablish constant activity during 5 min waiting time after breaking the seal. In case of AAV::ChR2: Increasing frequencies (5, 10, 20, 40, 80 Hz) of optogenetic pulses (473 nM) were applied to test for opto fidelity. In case of AAV::Arch, cells were subjected to an increasing ramp of 1 s depolarizing current pulses, each one accompanied by 400 ms of 20 Hz optogenetic pulses (563 nM) after 300 ms.

## Stereotaxic surgery

Surgeries were performed using a Model 1900 Stereotactic Alignment Instrument (David Kopf Instruments) and a Model 1911 stereotactic drill (David Kopf Instruments). For injections, a Nanoliter 2000 injector, driven by a Micro4 MicroSyringe Pump Controller (World Precision Instruments), was used. Needles for virus injection were pulled from 3.5 nl glass capillaries (World Precision Instruments) on a

Micropipette Puller (Model P-97, Sutter Instruments). The surgical protocol was adapted from *Athos and Storm, 2001*: Mice were deeply anesthetized in the stereotactic frame with isofluorane (1.7%, IsoFlo, Abbot Laboratories) and anesthesia was verified by testing deep plantar reflexes. Gentamicin ointment (Refobacin 3 mg/g, Merck) was used to protect the animals' eyes, and their body temperature was kept constant at 36°C using a heating pad. For optogenetic manipulation, one of the following viral constructs (see Key resources table) was bilaterally injected directly into the STN at –1.85 mm AP/ ±1.60 mm ML/–4.70 mm DV from bregma at a speed of 20 nl/min followed by a 5 min waiting period where the needle was kept in place in order to avoid leakage. Final injected volume was 100 nl in case of AAV::GFP and AAV::ChR2 groups, with 200 nl used for the rest of experimental groups. After the injection optic fibers (MFC_200/245-.053_5.0 mm_ZF1.25(G)_FLT from Doric Lenses) were implanted at –2.18 mm AP/±1.60 mm ML/–4.55 mm DV from bregma. For electrophysiological assessment, the animals were trained in GNG task till achieving performance and silicone electrodes (A1 × 16 poly2 50 × 375; Neuronexus) were implanted at STN (–1.85 mm AP/±1.60 mm ML/–4.70 mm DV from bregma) or SNr (–3.14 mm AP/±1.50 mm ML/–4.70 mm DV from bregma). Ground screws were mounted above the contralateral prefrontal cortex and cerebellum. All implants were fixed to the skull with dental cement (SuperBond C&B kit, Prestige Dental Products). Animals were given enrofloxacin (100 mg/ml, Baytril, Bayer Austria) and carprofen (Rimadyl, 50 mg/ml; Pfizer Austria) via drinking water for at least 7 days and were granted a resting period of at least 14 days before behavioral retraining commenced.

## Viral knockdown of mGlu4

To suppress mGlu4 expression in the STN, we constructed an AAV-based vector expressing GFP and miRNA-adapted shRNAs in the optimized miR-E backbone (under control of the SFFV promoter; AAV-SFFV-GFP-miR; ASGE), as described elsewhere (*Groessl et al., 2018*). Four independent shRNAs targeting mGlu4 (guide sequences: 5'-TAGTA) were designed based on optimized design rules (see Key resources table) and cloned into miR-E (*Fellmann et al., 2013*) and the mix was used to make viral preparation for surgical injection.

To test the knockdown efficiency, mouse neuroblastoma cell line (Neuro-2a, ATCC CCL-131) were transfected with the constructs mix or control plasmid using Lipofectamine 3000 according to manufacturer's instructions. Cells were harvested 72 hr after transfection, cell pellet was resuspended in extraction buffer consisting of 20 mM Tris–HCl (pH 7.5), 100 mM NaCl, 5 mM MgCl, 2 mM NaF, 10% glycerol, 1% NP40, 0.5 mM DTT, supplemented with protease inhibitor cocktail (Complete EDTA-free, Roche) and lysed on ice for 5 min. Input lysates and immunoprecipitates were resuspended in SDS sample buffer and heated to 95°C for 2 min. Total protein amount was assessed using Pierce BCA Protein Assay Kit according to manufacturer's instructions (Thermo Fisher Scientific). Pre-cast NuPAGE Novex 4–12% Bis-Tris midi gels (Invitrogen, XP04122BOX) were run in NuPAGE MOPS SDS running buffer (Thermo Fisher Scientific, NO0001). The protein samples were transferred onto a cellulose membrane using semi-wet transfer. The membranes were blotted with rabbit polyclonal antibody against mGlu4 (1:500 ab53088, Abcam) and mouse monoclonal anti-β-actin (1:5000 A5441; Sigma). The mGlu4 expression was assessed with ImageJ and normalized to β-actin signal for each sample.

## GNG task and performance criteria

The protocol was based on published material (*Gubner et al., 2010*; *McDonald et al., 1998*; *Moschak et al., 2012*). All animals were water deprived over-night to increase their motivation for a milk reward (10% condensed milk in water). The experiments were conducted in, in-house customized, Coulbourn behavioral testing boxes. For the experimental setup, a custom-built port with rectangular entrance in which the animal can insert the entire head was used. This port was equipped with a liquid delivery system and a blue/yellow (balanced) LED behind the port which can illuminate the port. Furthermore, it possessed two IR beams, first one to detect nose pokes (visits to port) and an additional one to detect licks at the liquid delivery system tube. The cage was cleaned with 70% ethanol before each mouse. Prior to GNG training, animals were habituated for one session in which the reward was dispensed in the port at variable intervals. Each reward occurrence was associated with a click sound that was then used throughout the entire behavioral experiment. The habituation session contained 60 reward deliveries without cue presentation, with house light on and separated by light off inter-trial interval (ITI) of 10 s.

Each trial of the GNG task consisted of four periods. The precue period of varying duration (9–24 s) was signaled by the house light turning on. Responses during this period were recorded but not reinforced. The variable timing was used to avoid predictive behavior toward the presentation of the cue. Any port visit during the last 3 s of the precue period terminated the trial (jump to ITI) in order to prevent false positive/false negative response counts upon cue presentation. Cue presentation followed the successful passing of the precue period and consisted of an auditory cue. The sounds chosen were a pulsed white noise for the Go trials and a 3 kHz pure tone for the NG trials, both set at 75 db intensity. The first visit to the response port, during the cue presentation, ended the cue and led to: (1) the reward delivery period (3 s), in case of a correct Go response; or (2) a jump to the ITI without reward delivery if a poke was detected during NG cue presentation (named false alarm, FA). Conversely, a successful NG trial was achieved by withholding poke responses in the port during the entire NG cue presentation period and resulted in reward delivery after cue end. In both correct responses, the reward was coupled with a 'click' sound and an LED light in the port and consisted of a 20 µl 10% milk solution. Licks were recorded for the duration of the entire trial. Each trial ended with a 10 s ITI during which the house light was off. Responses during this period were recorded but had no consequences.

Following habituation, animals started the first training phase of the GNG task. Here, only Go trials were used, where an animal had to deliver a port response during the 30 s Go cue presentation. Training occurred daily and consisted of 60 Go trials in each session or 40 min, whichever came first. After the animals reached a correct response rate of at least 60% and their performance was stable over the course of three consecutive sessions, the next training phase started.

In the second training phase, animals were introduced to the NG cue and cue presentation time was set to 10 s for both Go and NG trial types. Each session consistent in total of 60 trials (30× Go and 30× NG, random order) or 40 min, whichever came first. Animals were trained until they reached a correct response rate of at least 80% paired with an FA rate of maximum 45% and their performance was stable over the course of three consecutive sessions. Animals that reached these conditions further underwent stereotaxic surgery. After the post-operative rest period, the mice were retrained until they reached the performance criteria for the optogenetic and pharmacological experiments. For the shRNA experiment, animals were tested every second day for a total of seven sessions. Additionally, animals which on a non-treatment day performed less than 80% correct Go responses and/or made more than 60% of FA during NG trials were excluded.

Passing these criteria, animals were assigned to HI/LI groups based on 25th/75th percentile or median splits. This yielded individually consistent categorization of HI/LI animals, which was comparable across experiments (*Figure 6—figure supplement 7A*).

Behavioral data was processed using MATLAB programs (R2015b, MathWorks, Natick, MA), videos were analyzed in Ethovision XT 8 (Noldus Information Technology, Wageningen, The Netherlands), both data types were then merged and processed using custom-made Python scripts (Python 3.3) with the final analysis being done in GraphPad Prism (Version 7).

## Open-field test

Before each experiment, mice were allowed to habituate to the experimental room for at least 30 min prior to any testing.

The optogenetic manipulation was done with three 2 min laser off periods alternated by 2 min laser on periods in between (for laser set up, see Optogenetic manipulation). In case of mGlu4 PAM evaluation, the animals were first administered the drug or vehicle in a latin square design (see Pharmacology) and after a 30 min waiting period were placed in the arena and allowed to explore for 30 min. In both cases a 50 cm (width) × 50 cm (length) × 29.5 cm (height) arena was used, and video tracked using AnyMaze software (Stoelting). In the software, a 'center' zone was defined as a central square 25 cm × 25 cm in size, the rest being the 'border zone' (*Pliota et al., 2018*).

The open-field evaluation of shRNA knockdown for mGlu4 was done at the preclinical phenotyping facility of the Vienna Biocenter Core Facilities GmbH (VBCF). The animals were transferred to the facility 1 week prior to experiments and housed at a 14 hr light–10 hr dark cycle in IVC racks with access to food and water ad libitum. After placing in the open-field arena (50 cm (width) × 50 cm (length) × 29.5 cm (height)), mice were allowed to explore for 30 min and were video tracked using

TSE VideoMot 3D Version 7.01 software (https://www.tse-systems.com). In the software, a 'center' zone was defined as a central square 25 cm × 25 cm in size, the rest being the 'border zone'.

In all cases light conditions were about 300 lux in the center zone. After each trial, the apparatus was cleaned with water and 70% ethanol. Open-field experiments were performed in the morning (10:00 am–01:00 pm). The time spent in each zone, distance traveled, and number of center visits were recorded as readout parameters.

## mGlu4 PAM pharmacology

4-((*E*)-styryl)-pyrimidin-2-ylamine (mGlu4 PAM, Cmp 11) was synthesized at Boehringer Ingelheim, Germany. The compound was dissolved in one volume of 0.1% Tween-80 (v/v) and nine volumes of 0.5% Natrosol and administered orally at 10 ml/kg, for a final dose of 80 mg/kg, 30 min before testing. The dose was chosen based on previous studies (*East et al., 2010*; *Isherwood, 2017*).

## In vivo electrophysiology

After the surgery and recovery period, animals were handled and habituated to the recording room for several days prior experimental recordings. Electrodes were connected via an Omnetics connector to a 16-channel unity-gain headstage (Plexon) and the animal was left in the homecage for 10 min thereafter. The headstage was connected to a pre-amplifier where the signal was band-pass filtered (3 Hz – 8 kHz) and amplified. Neural activity was digitized at 40 kHz and highpass-filtered for spikes (800 Hz) and LFPs (3–200 Hz) for offline analysis. All recording sessions per mouse were merged. In general, three no treatment sessions, two vehicle, and two mGlu4 PAM sessions were concatenated together and split accordingly to type after unit sorting. Single units were sorted manually with Offline Sorter v4 (OFS, Plexon) in 3D PC feature space on unsorted waveforms (*Figure 2—figure supplement 2A*) and declared a single unit if the spike cluster was separable from noise and other clusters and no refractory period infringements were present. To avoid multi-sampling of single units, cross-correlograms of units from adjacent channels were inspected for co-firing and respective units removed from analysis. Simultaneously to neuronal data acquisition, animal behavior was recorded using both the sensor state (as described in the GNG task section) and image, with a video camera located on top of the cage, using CinePlex Studio. Video analysis and synchronization with neuronal recordings was then performed using CinePlex Editor. For the extraction of the immobility episodes, the motion measure data was averaged with the moving average window of 2 s, with the minimum duration of an immobility state set at 1 s (threshold 80). Episodes less than 0.5 s apart were merged. Finally, Neuroexplorer 5 was used to reconstruct the behavioral paradigm and treatment sessions.

Unit activity is reported either as frequency (Hz) or normalized activity (Z-score transformation). Units with less than 20 spikes across the entire sessions were removed from further analysis. For event-related analysis (i.e., poke onset), the activity of each unit was transformed to Z-scores using the mean and SD of session-wide firing rate (250 ms bin windows) and moving-window smoothened using 1.5 s Gaussian. A unit was considered to be event-related if its activity in any of the post-onset bins (0 s to +1 s) was significantly different from mean baseline (–2 s to –1 s) activity across all trials using one-sample t-test. To show significant change of population firing from session-wide mean, a two-sided one-sample t-test against 0 for each time bin was used and was further corrected for multiple comparisons using cluster-based permutation testing on contiguously significant bins using 5000 iterations (*Maris and Oostenveld, 2007*). In order to test if given responsive population on given event reaches significance, the null distribution of number of responsive units was built by performing 5000 iterations of randomly shifting the event onsets and performing the even-related analysis as previously described. The observed number of units was then compared to the obtained shuffled distribution and was considered significant if the observed number lied within the top 10% of the distribution. Low-dimensional representations for visualizing changing of population dynamics over time were constructed using PCA on trial averaged unit-based peri-event time course for the entire population using selected features (shown on the particular graph) and time window between –5 s and 5 s (*Jolliffe and Cadima, 2016*).

Three mice were implanted for STN. Each animal was recorded for two to four independent non-treatment sessions resulting in a total of 74 units (average eight units per session), two sessions under vehicle resulting in a total of 27 units (average five units per session) and three sessions under mGlu4 PAM resulting in a total of 34 units (average of four units per session). Session-averaged precue

response rates for each animal used for the STN electrophysiological analysis are presented in *Figure 6—figure supplement 7B*. Two animals were implanted for SNr. Each animal was recorded for four independent non-treatment sessions resulting in a total of 41 units (average five units per session), one session under vehicle resulting in a total of 11 units (average five units per session) and one session under mGlu4 PAM resulting in a total of seven units (average of three units per session).

## Optogenetic manipulation

Animals injected with optogenetic AAV for later neuronal modulation during behavior underwent habituation for attaching a fiber-optic patch cord (Doric lenses) onto the implanted optical fibers. For ChR2 activation, laser trains of blue light (473 nm) consisting of 20 ms pulses with a frequency of 20 Hz (if not noted otherwise) were delivered at an intensity of 1–1.5 mW at the fiber tip, unless stated otherwise. For Arch-mediated silencing, laser trains of constant yellow light (568 nm) were delivered at an intensity of 8–10 mW. Intensity of all laser stimulations was measured before every experiment at the tip of the optic fiber via Power Meter (Thorlabs, PM100D). Laser stimulation was controlled by MATLAB scripts during GNG experiments and by Arduino boards running customized scripts executed by Any-maze software (Stoelting) during open-field test.

## mGlu4 PAM pharmacology

4-((*E*)-styryl)-pyrimidin-2-ylamine (mGlu4 PAM, Cmp 11) was synthesized at Boehringer Ingelheim, Germany. The compound was dissolved in one volume of 0.1% Tween-80 (v/v) and nine volumes of 0.5% Natrosol and administered orally at 10 ml/kg, for a final dose of 80 mg/kg, 30 min before testing. The dose was chosen based on previous studies (*East et al., 2010*; *Isherwood, 2017*).

## Resting-state functional magnetic resonance imaging

Animals were administered vehicle or 80 mg/kg mGlu4 PAM (see Pharmacology) and were left undisturbed for 30 min prior any MRI measurements. MRI was performed on a 15.2T Bruker system (Bruker BioSpec, Ettlingen Germany) with a four-channel phase array coil for mouse heads (Bruker, Biospec). Prior to imaging all mice were anesthetized with 4% isoflurane, while care was taken to adjust the isoflurane levels immediately so that respiration did not go below 100 beats per minute (bpm) at any time. During imaging, respiration was kept between 110 and 140 bpm. For the rs-fMRI study, single shot echo planar imaging sequence with spin echo readout was used (TR = 2000 ms, TE = 19.7 ms, FOV = 16 × 16 mm$^2$, voxel size = 250 × 250 μm$^2$, 30 slices 0.5 mm thick, 1 average, 240 repetitions, 8 min total imaging time). Following resting-state scan, a high-resolution T1-weighted anatomical scan was acquired using gradient echo sequence (TR = 500 ms, TE = 3 ms, FOV = 16 × 16 mm$^2$, voxel size = 125 × 125 μm$^2$, 30 slices 0.5 mm thick, 4 averages).

rs-fMRI data were first bias-field corrected using N4ITK algorithm (*Tustison et al., 2010Sled et al., 1998*,). Pre-processing was done using the Data Processing Assistant for Resting-state fMRI Advanced Edition (DPARSF-A) toolbox, which is part of the Data Processing and Analysis of Brain Imaging (DPABI) toolbox Version 3.1 (http://rfmri.org/dpabi, *Chao-Gan and Yu-Feng, 2010*). The first 10 volumes were removed from each resting-state dataset. Data were processed in series of steps that included slice-timing correction, realignment, co-registration, normalization, and segmentation using in-house created mouse masks for cerebrospinal fluid (CSF), white matter (WM), and gray matter (GM). Nuisance covariates related to the motion were regressed out using the Friston 24-parameter model (*Friston et al., 1996*). In addition, WM and CSF mean time-series were used as nuisance regressors in the general linear model to reduce the influence of physiological noise (*Margulies et al., 2007*). Global signal regression was used (*Saad et al., 2012*). Data were smoothed spatially with a 2.4 pixel full-width half-maximum Gaussian kernel. All data were co-registered to the in-house generated mouse atlas (with 102 distinct brain regions). Data were corrected for multiple comparisons using Gaussian random field theory multiple comparison correction (voxel-level p-value = 0.05, cluster-level p-value = 0.05).

Functional connectivity (FC) patterns were compared between each group, by extracting the mean time-series BOLD signal of each of 102 brain regions. In order to investigate the general effect of impulsivity across all annotated brain regions, mean FC correlations were calculated for each animal under vehicle treatment, and the mean correlation matrices for each impulsivity group were subtracted from each other. Next, for each brain node, a one-sample t-test was performed against 0 if no difference

on impulsivity levels, using Bonferroni for multiple correction. To assess the interaction effect between mGlu4 PAM treatment and impulsivity levels, FC correlation matrices were calculated for each animal under vehicle or compound treatment as before, followed, by two-way repeated-measures ANOVA on each of the respective brain node, using Bonferroni for multiple correction.

## Histological analysis

To verify virus expression (see Key resources table for viral constructs) and correct locations of optical fiber tips and cannulae, animals were sacrificed using a mixture of 10 mg/ml ketamine (OGRIS Pharma) and 1 mg/ml medetomidine hydrochloride (Domitor, ORION Pharma) in 1× PBS and transcardially perfused with 40 ml of cold 1× PBS/heparine followed by 40 ml of 4% (PFA) in 1× PBS. Brains were immediately removed, post-fixed in 4% PFA at 4°C overnight, transferred to 30% sucrose solution for 24 hr and subsequently frozen in Tissue Tech O.C.T. on dry ice and stored at –80°C until sectioning. Coronal cryosections were cut at 20 µm thickness. Selected sections were counterstained with DAPI and mounted in Fluorescence Mounting Medium (Dako, S3023). Whole slides were then scanned using an automated widefield microscope (Pannoramic 250 Flash, 3D HISTECH Ltd.). Once images were acquired, regions of interest (ROIs) were marked by hand within Pannoramic Viewer (3D HISTECH Ltd). Expression of viral constructs and location of optical fiber tips/cannulae were then assessed for correct targeting (*Figure 3—figure supplement 1A*).

Viral expression was quantified using a custom ImageJ script, using an overlay of the appropriate sections from all animals within the experimental group and averaging the signal.

## In situ hybridization

The mGlu4 expression analysis was done on cryosections with tissue prepared as described for the histological analysis extracted from GNG trained animals. The multiplexed in situ hybridization staining was done using the RNAscope system (RNAscope Multiplex Fluorescent v2 kit Cat no. 323100, Advanced Cell Diagnostics) according to manufacturer's protocol. The tissue was co-hybridized with proprietary probes against *Grm4* (Cat no. 480991), probe mix against *Slc17a6+ Slc17* a7 (Cat no. 416631 and 319171) followed by differential fluorescence tagging (TSA Cy 3, 5, TMR Fluorescein Evaluation kit Cat no. NEL 760001KT, Perkin Elmer). Slides were imaged as described in the Histological analysis section. Images were processed using ImageJ, quantifying the total number of mGlu4 foci, and normalizing each ROI to the total number of detected nuclei using DAPI staining. For each brain region, ROIs at least two sections were selected for left and right hemispheres, averaged, and treated as independent samples. mGlu4 expression quantification was performed on these ROIs by automated analyses based on custom ImageJ scripts. The signal for each ROI was normalized to the average signal in the LI group for regional analyses. For cellular analyses, perisomatic cellular signals and cell type (VGlut) were identified using custom scripts. Cells were defined as VGlut$^+$ if their perisomatic signal was above the ROI section average for a conservative classification of VGlut$^+$ glutamatergic projection neurons.

## Statistical data analysis

Sample sizes are in line with estimates derived from previous experiments, using G*Power Version 3.1.9.6. For behavioral experiments, target sample size was in the range of 6–10 animals (effect size 0.45, *Groessl et al., 2018*). Animals were assigned randomly to all experimental cohorts. The behavioral experimenter was blind to the treatment of respective groups, wherever possible. All behavioral and data analyses were carried out blinded and using automated computational pipelines, wherever applicable. Establishing of the behavioral assay, neural recordings, and circuit manipulation were performed in independent experiments, with separate animal cohorts, wherever applicable. Basic behavior was replicated across experiments for control groups. Optogenetic manipulations were replicated on separate experiments and cohorts (biological replicates). All behavioral statistics were performed in GraphPad Prism (Version 7), unless otherwise indicated, and all statistical tests used are indicated in the figure legends. Experimental designs with one categorical independent variable were assessed by Shapiro-Wilk normality tests. If normality test passed, parametric statistics (t-test, one-way ANOVA) were applied. In case of non-normal distributions, non-parametric statics (Mann-Whitney U-test and Wilcoxon signed rank test) were planned. Experimental designs with two categorical independent variables were assumed to be normal and analyzed by two-way (optional: repeated) ANOVA

without formally testing normality, followed by Sidak's multiple comparison test. All significance levels are given as two-sided and were corrected for multiple comparisons, wherever applicable. The HI/LI split for pharmacological and shRNAi knockdown experiments was done by pooling the non-treatment data from both groups and performing a median split, treating the grouping as factor for subsequent statistical analysis, as described elsewhere (*Iacobucci et al., 2015*). For all behavioral cohorts, a Grubb's outlier test was performed at alpha = 0.0001 for the main parameters in the GNG task (precue poke rate, FA, and correct Go responses). In case of comparing distributions (i.e., precue response rate, latency to poke), a two-sample Kolmogorov-Smirnov test was used where data from all trials and all animals within a given group were pooled for analysis ( *Figure 1—figure supplement 1—source data 1*, *Figure 3—source data 1*, *Figure 4—figure supplement 1—source data 1* and *Figure 6—figure supplement 3—source data 1*). The multivariate comparison of behavioral experiments (i.e., precue response rate and % of NG trials with FA) were assessed by multivariate ANOVA (with repeated measures if applicable) followed by ANOVA in case of significance for the given measure. Data for rs-fMRI were analyzed as reported in the Resting-state functional magnetic resonance imaging section, one-sample t-test, and two-way ANOVA results are reported in *Figure 1—source data 1* and *Figure 4—figure supplement 2—source data 1*, respectively. Omnibus significance values were rounded up for values $p < 0.0001$. Post hoc significance values were rounded up and given as * for values $p < 0.05$, ** for values $p < 0.01$, *** for values $p < 0.001$, and **** for values $p < 0.0001$; where no significance was made explicit, the test did not reach a significance level of $p < 0.05$. Unless stated otherwise, data are shown as mean ± SEM.

## Data exclusion

For behavioral experiments, animals were excluded based on the following criteria. Note that some animals might fit more than one exclusion criteria.

Histology exclusion (all sessions): Animals were removed after histological evaluation due to incorrect targeting or low viral expression.

Technical exclusion (per session, i.e., laser in precue and/or cue-sessions, vehicle and/or treatment sessions): software malfunction, excessive coiling of the laser fibers or the loss of fibers during the task for optogenetic cohorts; malaise due to gavaging for the mGlu4 vehicle/PAM cohorts.

Non-performance exclusion (per session): all cases of animals not reaching fixed thresholds for correct Go of 80% and FA of 45% for 3 consecutive prior days (baseline) (post-surgery, if applicable).

Statistical outlier exclusion (per parameter): data points excluded for statistical power after stratification by Grubb's outlier removal for individual parameters using conservative settings (alpha = 0.0001), followed by mixed-model ANOVAs where applicable.

Applying these criteria stratified the cohorts as follows.

Optogenetic cohorts: 2 out of 20 AAV::GFP, 2 out of 15 AAV::ChR2, and 6 out of 33 AAV::Arch did not reach sufficient virus expression and/or missed injection targets and were excluded.

In the GNG task during laser manipulation in the precue period, from the remaining animals, 4 AAV::GFP, 2 AAV::ChR2, and 8 AAV::Arch animals were excluded for technical reasons and 1 AAV::GFP and 2 AAV::Arch animals were excluded for non-performance.

In the GNG task during laser manipulation in the cue period, from the remaining animals, 3 AAV::GFP, 3 AAV::ChR2, and 7 AAV::Arch animals were excluded for technical reasons and 1 AAV::GFP, 3 AAV::ChR2, and 5 AAV::Arch animals were excluded for non-performance. mirE-cohorts: 2 out of 19 animals for mirE-control and 3 out of 20 animals for mirE-mGlu4 did not reach sufficient virus expression and/or missed injection targets and were excluded. In the GNG task, from the remaining animals, one animal from mirE-Renilla and one animal from mirE-mGlu4 were excluded for non-performance.

mGlu4 PAM cohorts: In the GNG task 4 out of 31 animals from mGlu4 PAM and 2 out of 31 animals from vehicle treatment groups were excluded due to malaise. Further, from the remaining animals, one animal from the mGlu4 PAM group was excluded for non-performance in the task. Finally, two animals from the HI/precue and LI/Go parameter were excluded as statistical outlier (Grubb's).

## Behavior scripts

Code for assessing behavioral parameters. MATLAB and Python scripts to extract all animal behavioral data from sensor data from the cage/behavior operating software, such as precue response rate, % of

correct Go responses. Used to generate main figures: *Figure 1B and C*; *Figure 3A–D*; *Figures 4A–C and 6C–E*.

## Electrophysiology scripts

Code for post-processing in vivo electrophysiological data. A set of R scripts to process and combine the electrophysiological data exported from Neuroexplorer for further analysis and visualization, with additional options to perform cluster permutation statistical tests. Used to generate main figures: *Figures 2A–E and 5A–C*.

## fMRI data plus scripts

Data and code for fMRI. A set of R scripts to process the time-series BOLD signal from the fMRI data-sets exported from DPABI, applied to generate correlation matrices, perform one-sample t-test and ANOVAs. Used to generate main figures: *Figures 1D, E and 4D–E*.

## Histology scripts

Code for cell counting from histological images. A custom-made ImageJ script to evaluate in situ hybridization signals from images obtained using an automated widefield microscope. Used to generate main figures: *Figure 6B*.

## Acknowledgements

The Preclinical Phenotyping Facility (Vienna Biocenter Core Facilities; VBCF, Austria) performed open-field behavioral experiments on the shRNA groups. Florian Grössl (IMP) contributed to the in vitro validation of optogenetic manipulations. We thank the VBCF Histopathology department (Michaela Zeba) and the IMP/IMBA BioOptics facility (Thomas Lendl) for their help with imaging and histological analysis of brain sections. Special acknowledgments go to Manuel Pasieka (VBCF Scientific Computing) for designing and creating the operating software for the behavioral boxes and data extraction algorithms for the Go/No-Go task and to Sophia Ulonska and Katja Bühler (Zentrum für Virtual Reality und Visualisierung, VRVis) for contributing to neuronal data analysis. We thank the IMP/IMBA Comparative Medicine Core Facility for colony management; Aline Romer and Florian Hofer for contributing to behavioral experiments; Nathan Lawless (Boehringer Ingelheim) for contributing the meta-analyses and database searches, and Bastian Hengerer, Stefan Just, and Moritz von Heimendahl (Boehringer Ingelheim) for their comments and advice. Finally, Johannes Zuber designed the shRNA constructs, Barbara Werner cloned and prepared the shRNA viruses and Maciej Żaczek assisted with immunoblotting. WH was supported by a grant from the European Community's Seventh Framework Programme (FP/2007–2013), an ERC grant (agreement no 311701), the IMP, Boehringer Ingelheim, and the Austrian Research Promotion Agency (FFG). LP and AC were supported by a grant from Boehringer Ingelheim. LP is a member of the Boehringer Ingelheim Discovery Research global post doc programme. The Preclinical Phenotyping Facility acknowledges funding from the Austrian Federal Ministry of Science, Research and Economy, and the City of Vienna. We thank Life Science Editors for scientific editing of the manuscript.

## Additional information

### Competing interests

Anton Pekcec, Janet R Nicholson: is affiliated with Boehringer Ingelheim Pharma GmbH and Co. The author has no competing and/or financial interests to declare. The other authors declare that no competing interests exist.

### Funding

| Funder | Grant reference number | Author |
|---|---|---|
| H2020 European Research Council | 311701 | Wulf Haubensak |

| Funder | Grant reference number | Author |
|---|---|---|
| Boehringer Ingelheim | | Wulf Haubensak |
| Österreichische Forschungsförderungsgesellschaft | | Wulf Haubensak |

The funders had no role in study design, data collection and interpretation, or the decision to submit the work for publication.

### Author contributions

Lukasz Piszczek, Conceptualization, Data curation, Formal analysis, Investigation, Methodology, Project administration, Software, Validation, Visualization, Writing – original draft, Writing – review and editing; Andreea Constantinescu, Conceptualization, Investigation, Methodology, Validation, Writing – original draft; Dominic Kargl, Conceptualization, Data curation, Formal analysis, Investigation, Methodology, Visualization, Writing – review and editing; Jelena Lazovic, Formal analysis, Investigation, Methodology; Anton Pekcec, Conceptualization, Supervision; Janet R Nicholson, Supervision; Wulf Haubensak, Conceptualization, Funding acquisition, Project administration, Supervision, Writing – original draft, Writing – review and editing

### Author ORCIDs

Lukasz Piszczek (ID) http://orcid.org/0000-0003-2017-8853
Andreea Constantinescu (ID) http://orcid.org/0000-0002-3081-0755
Dominic Kargl (ID) http://orcid.org/0000-0001-7206-1708
Wulf Haubensak (ID) http://orcid.org/0000-0002-2034-9184

### Ethics

Animal procedures were performed in accordance with institutional guidelines and were approved by the 4 respective Austrian (BGBl nr. 501/1988, idF BGBl I no. 162/2005) and European authorities (Directive 86/609/EEC of 24 November 1986, European Community) and covered by the license M58/002220/2011/9.

### Decision letter and Author response

Decision letter https://doi.org/10.7554/eLife.62123.sa1
Author response https://doi.org/10.7554/eLife.62123.sa2

## Additional files

### Supplementary files

• Transparent reporting form
• Source code 1. Code for assessing behavioral parameters.
• Source code 2. Code for post-processing in vivo electrophysiological data.
• Source code 3. Data and code for fMRI.
• Source code 4. Code for cell counting from histological images.

### Data availability

All data generated or analysed during this study are included in the manuscript and supporting files. The custom scripts for the analyses of fMRI-, behavior-, electrophysiology-, and histology-related datasets are enclosed as source code files.

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
