## [Editor Report]

Piszczek et al., probed mGluR4 modulation of impulsivity at the systems and molecular level using resting fMRI, in vivo/ex vivo electrophysiology, pharmacological, optogenetic, and genetic manipulations in mice. Authors link behavioral trait variation to an mGluR4-dependent mechanism in the subthalamic nucleus. Overall, the identification of mGlu4 as a marker for high trait impulsivity reveals a novel potential therapeutic.

---

## [Decision Letter]

**Decision letter after peer review:**

Thank you for submitting your article "Dissociation of impulsive traits by subthalamic metabotropic glutamate receptor 4" for consideration by *eLife*. Your article has been reviewed by 3 peer reviewers, including Alicia Izquierdo as the Reviewing Editor and Reviewer #1, and the evaluation has been overseen by Kate Wassum as the Senior Editor. The following individuals involved in review of your submission have agreed to reveal their identity: David Jentsch (Reviewer #2); M. Foster Olive (Reviewer #3).

The reviewers have discussed the reviews with one another and the Reviewing Editor has drafted this decision to help you prepare a revised submission.

Summary:

Piszczek et al., probed metabotropic glutamatergic (mGluR4) modulation of impulsivity at the systems and molecular level using a powerful toolkit: resting stage fMRI, in vivo/ex vivo electrophysiology, pharmacological, optogenetic, and genetic manipulations. Authors focused on the subthalamic nucleus (STN) and on natural variation in trait impulsivity, separating it from motor responding, using an adapted trial-cued Go/NoGo task. STN showed the highest functional connectivity fMRI measures in low impulsive animals compared to high impulsive animals. Authors also found that STN encodes impulsivity features in precue pokes, but not Go responses, and further, that STN population activity gatekeeps precue pokes. Additionally, viral mGluR4 knockdown in STN decreased impulsivity, and mGluR4 manipulation via optogenetic inhibition affected impulsive behavior without affecting global motor function. The overall focus on mGluR4 in the STN appeared clearly to be an a priori objective, rather than emerging from the discovery efforts described herein. Overall reviewers found this to be a thorough, technically strong study. However, there was strong consensus about a need for greater statistical transparency, particularly about sample size, and clarity of data presentation-especially for the rs-fMRI and electrophysiological recording aspects of the work.

Essential revisions:

– The abstract is rather brief and does not fully describe all of the findings. Specifically, optogenetic, pharmacology, and e-phys results are not summarized.

– A concern raised by reviewers has to do with sample sizes and the corrections for multiple comparisons utilized. Group sizes are low (n=3) in some aspects of the study, which may lead to statistical underpowering and false negative results. In particular, the rs-fMRI study involved a total of 10 subjects divided in two groups. At least 204 correlations were computed given the BOLD time series (2 hemispheres x 102 seed regions). The authors should explain a bit better why they used an approach wherein they subtracted the correlation values in the HI and LI mice from one another and then compared them to 0 using a one-sample t-test, rather than simply reporting the p value associated with the contrast relative to the corrected p value threshold. I note, based upon Suppl Data 3 table, that the main effect of Impulsivity was not significant for any brain region, including the STN (though I also recognize this is from the full 2 x 2 analysis with active and control treatments, so not the same as the simple contrast under vehicle only conditions). Please provide more detailed justification for the approach used.

– The initial rs-fMRI study examined the functional connectivity between 102 regions of interest (ROI) the rest of the brain in HI and LI mice. More rationale for this approach should be provided. It is well understood, from studies of both human and animal subjects that systematic whole brain analyses of functional connectivity consistently reveal evidence for distinct functionally interactive networks (e.g., the default mode network, the dorsal attention network, etc.) in which specific brain regions show patterns of strong functional connectivity with other structures in their network, but not with structures outside of their network. This is also apparent when one uses a seed-based approach – starting with the time series in one ROI (say, the STN), correlation in the BOLD time series will vary from brain structure to brain structure, reflecting the fact that some structures are differentially strongly connected to various other brain regions. It is unclear, therefore, what patterns of correlation in the fMRI BOLD time series between one structure and all other structures really means, biologically, and why this approach was used to analyze the dataset generated here.

– From the analyses of the mGluR4 PAM rs-fMRI study (Suppl Data 3) it appears that the most consistent set of observations (and perhaps largest effects) relate to a dramatic main effect of treatment across a broad range of brain regions. This is not addressed anywhere in the text but seems to be an important context in which to understand the treatment x impulsivity interactions observed, including in the STN.

– Authors report they recorded LFPs, but do not show these data in frequency bands; please show these. The unit recording component of this collection of studies is particularly intriguing and represents one of the more novel aspects of the work. One question that arises, however, stems from the fMRI experiments. Were the 3 mice in this study LI or HI? What is the behavioral context for these neuronal responses? If these mice were LI, what neuronal responses might be found in HI, or vice versa? If they were average impulsivity mice, how might these observations extend to HI or LI subjects? Please clarify.

– A similar question to that above extends to the optogenetic experiments. The performance of the animals in the control (no laser) conditions appears to be quantitatively similar to the HI mice in the MRI experiment. Were mice selected for HI phenotype? If they were a random group, this raises the question of reproducibility of the LI vs. HI phenotype.

– Only a single dose of the mGlu4 PAM was administered, which limits interpretation of the findings. In addition, the drug was administered systemically, where it could interact with mGlu4 receptors in a number of brain regions to produce its effects. Site-specific microinjections would have been more informative. Authors are encouraged to address this limitation in the Discussion.

– Related to the pharmacology aspect and clarifications in the Discussion: the conceptual novelty may be a bit overstated as others have shown before that there is an interaction between D2 receptor antagonism and allosteric modulation of mGluR4 in impulsivity (using traditional pharmacology and a 5-Choice Serial Reaction Time Task). Motor and choice impulsivity (akin to what the authors refer to as trait impulsivity) has been previously dissociated via the mGluR4 receptor (Isherwood et al., 2017 Neuropharm). So there is precedent for its potential involvement in impulsivity albeit using mostly classic systemic administration methods. Indeed, the role of mGluR4 may likely be via D2R, which is an older story. It is recommended that authors incorporate a consideration of this into their Intro and/or Discussion.

– The authors indicate that they intentionally sought to examine individual differences in these impulsivity traits in an isogenic strain in order to study them in the "same genetic and neuroanatomical context." This statement is somewhat perplexing as the largest source of individual differences in brain mechanisms recruited during go/no-go performance in human subjects is genetic (Anokhin et al., Int J Psychophysiol. 2017 May;115:112-124), as it is for impulsivity, more broadly. Thus, the mechanisms underlying inter-individual variation in impulsive behavior in the isogentic strain is explicitly not the same as the major driving factor influencing this in humans, potentially compromising the translational value of the findings. More discussion of this point is warranted, as are the rationale for, and limitations of, using an only male subject pool.

– There appears to be some overflow of virus expression into peri-STN regions such as the zona incerta, which the authors attribute to filling of neurites and/or projections. Higher magnification images showing that expression in these regions is confined to fibers and not cell bodies are needed to demonstrate this. This is especially important when stimulating with red light since it is capable of penetrating further into tissue than blue light.

– The mGlu4 hybridization signal is not exceptionally strong, nor is it concentrated in cell soma of vGluT positive neurons. Only a few red pixels can be observed, and some message signal appears to be extracellular or on fibers that are not abelled. A more convincing demonstration of mGlu4 expression in vGluT positive neurons is needed.

– Both the behavioral data from the mGluR4 PAM study and the mGluR4 knockdown study involved separate ANOVA in the HI and LI groups. This is not justified. An omnibus ANOVA with impulsivity trait group as a factor should be conducted, and only in the case of a relevant interaction should the groups be analyzed separately.

– In order to fully understand the effects of mGluR4 knockdown on behavior in HI and LI mice, molecular data on mGluR4 protein in HI and LI mice exposed to the active or control virus would be very helpful.

– A nontrivial 11 AAV::Arch animals were excluded for non-performance/technical issues. Can authors provide more information about this? Non-performance is very different than technical issues as an exclusion criterion. Please justify.

– For optogenetics studies, viral vectors used incorporated a pan-neuronal promoter (hSyn). Is there a reason why the authors did not choose a more selective cell-type specific promoter, say for excitatory neurons? Please clarify.

– Can the authors cite prior studies by other groups that have used similar behavioral splits based on distance travelled in the GNG task, licks, etc.?

---

## [Author Response]

Essential revisions:– The abstract is rather brief and does not fully describe all of the findings. Specifically, optogenetic, pharmacology, and e-phys results are not summarized.

We thank the reviewer for pointing this out.

We have amended the abstract to include the optogenetic, pharmacology, and e-phys findings.

– A concern raised by reviewers has to do with sample sizes and the corrections for multiple comparisons utilized. Group sizes are low (n=3) in some aspects of the study, which may lead to statistical underpowering and false negative results. In particular, the rs-fMRI study involved a total of 10 subjects divided in two groups. At least 204 correlations were computed given the BOLD time series (2 hemispheres x 102 seed regions). The authors should explain a bit better why they used an approach wherein they subtracted the correlation values in the HI and LI mice from one another and then compared them to 0 using a one-sample t-test, rather than simply reporting the p value associated with the contrast relative to the corrected p value threshold. I note, based upon Suppl Data 3 table, that the main effect of Impulsivity was not significant for any brain region, including the STN (though I also recognize this is from the full 2 x 2 analysis with active and control treatments, so not the same as the simple contrast under vehicle only conditions). Please provide more detailed justification for the approach used.

We thank the reviewer for the suggestion to provide more detailed rationale and description of fMRI design and analyses.

The cohort size that we used for our HI/LI comparison and mGlu4 x HI/LI interaction was limited by behavioral handling and fMRI operating time and costs. To extract meaningful data under these conditions, we took into account the following considerations.

First, the foremost goal of the fMRI experiments was to identify the most highly ranked nodes that control trait impulsivity. For this purpose, the fMRI survey was not intended to be exhaustive and, therefore, will contain false negative results. An exhaustive screen would require a larger sample size, but our low-powered screen should reliably identify the top-ranked nodes, despite the rather small sample sizes.

Second, the rather small cohorts demanded workflows that increase the power of detecting significant effects with the available sample sizes. One way to increase power is by omitting corrections for multiple comparisons (Ding et al., 2014; Filipello et al., 2018; Komaki et al., 2016; Liu et al., 2020; Sadaghiani et al., 2015; Wandres et al., 2021). We used this approach, therefore, which we believe is valid since we were primarily interested in ranking hot spots to capture the most prominent nodes, as shown for significant effects on the STN; we chose rank order (for both the HI/LI comparison and mGlu4 x HI/LI interaction) rather than directly reporting statistical significance *per se*. This is similar to published analysis pipelines for studies of both rodents (Bero et al., 2012; Cruces-Solis et al., 2020; Filipello et al., 2018; Liu et al., 2020; Tsurugizawa et al., 2020) and human subjects (Gohel and Biswal, 2015; Sohn et al., 2017), also for similar cohort sizes to ours (Egimendia et al., 2019; Sirmpilatze et al., 2019).

Third, for the HI/LI comparison we presented z-scored subtracted correlations as they report ranked measures more accessibly and effect size more intuitively than simple unidirectional p values. Likewise, for the mGlu4 x HI/LI interaction, we found z-scored (unidirectional) F-values, rather than p values of these comparisons *per se*, provided the most suitable measure to establish a rank order based on effect size. Specifically, we used an analysis pipeline similar to those published by others (Liska et al., 2015; Mennes et al., 2012; Rashid et al., 2019) (Liska et al., 2015; Mennes et al., 2012; Rashid et al., 2019), in which we collapsed the group-wise correlation matrix for a given brain region, collecting all of the correlations from both hemispheres into one vector. For the brain region ranking we subtracted such vectors for HI and LI groups from each other for each brain region separately, and then calculated the one-sample t-test on the difference between the groups. This approach has the advantage of minimizing the variance between animals – a critical factor in a low n experimental design – and thus increases power for our node ranking. Given the low power setting of our study, we felt that ranks are best represented by effect sizes (and only filtered by p values, rather than basing the ranks by p-values per se). We believe that a similar rank order would result from using p value.

For the mGlu4 x HI/LI interaction, we performed a two-way repeated measures ANOVA across rows, on individual matrices to decrease animal-to-animal variability and increase the chance of finding the most promising nodes. As we found several interactions in these analyses (including one for STN), the (absence of) treatment or trait main effects are difficult to interpret as such.

Thus, we believe that HI/LI comparison and mGlu4 x HI/LI interactions are best represented by z-scored differences in correlation values and F interaction terms, because they represent the most direct measure of effect sizes and ranks, which we then filtered for significance (Figure 1Ei-iii; Figure 4Ei-iii).

We have included a statement to convey these considerations in the Results section (Lines 136-156 and 331-336).

– The initial rs-fMRI study examined the functional connectivity between 102 regions of interest (ROI) the rest of the brain in HI and LI mice. More rationale for this approach should be provided. It is well understood, from studies of both human and animal subjects that systematic whole brain analyses of functional connectivity consistently reveal evidence for distinct functionally interactive networks (e.g., the default mode network, the dorsal attention network, etc.) in which specific brain regions show patterns of strong functional connectivity with other structures in their network, but not with structures outside of their network. This is also apparent when one uses a seed-based approach – starting with the time series in one ROI (say, the STN), correlation in the BOLD time series will vary from brain structure to brain structure, reflecting the fact that some structures are differentially strongly connected to various other brain regions. It is unclear, therefore, what patterns of correlation in the fMRI BOLD time series between one structure and all other structures really means, biologically, and why this approach was used to analyze the dataset generated here.

We thank the reviewer for pointing out the need for clarification of our node (vs. edge) centered approach.

The fMRI design ranked, seed-by-seed, the nodes that were most affected by the trait in the context of the brain network. This approach does not define the specific subnetworks these nodes might be embedded in (this will be the focus of a future study). We believe that our strategy, using a non-binarized node strength, is the best way to identify hotspots without referring to specific subnetworks. This strategy has been used by others to create similar node rankings for the investigated phenomena (Bero et al., 2012; Cruces-Solis et al., 2020; Filipello et al., 2018; Hsu et al., 2012; Liu et al., 2020; Rosenberg et al., 2015; Tsurugizawa et al., 2020). Our experimental strategy was to find hot-spots where the traits of interest are expressed by looking for highly modulated nodes. The effects on node-wise correlation is the most straightforward way of ranking node-wise effect sizes across the brain. This omnibus change in functional connectivity indicates a role of such nodes in organizing network dynamics, and their ability to influence the state not only of their structural neighbors but also of peripheral nodes. We suggest that this decrease of STN functional connectivity in HI animals reflects an overall reduction of this coupling and information transfer to the rest of the brain.

As mentioned above, this approach is limited since it does not yet address larger network effects and changes at the mesoscopic scale across multiple brain regions; however, the goal of our strategy was to get an entry point for investigating such functional subnetworks (STN-connected and beyond). Thus, we agree that a study on larger subnetwork connectivity is warranted in the future to find the specific STN connections that were modulated in our current study.

We have amended the manuscript to convey better these considerations (Lines 136-156 and 331-336).

– From the analyses of the mGluR4 PAM rs-fMRI study (Suppl Data 3) it appears that the most consistent set of observations (and perhaps largest effects) relate to a dramatic main effect of treatment across a broad range of brain regions. This is not addressed anywhere in the text but seems to be an important context in which to understand the treatment x impulsivity interactions observed, including in the STN.

We thank the reviewer for this suggestion to comment on this in the text.

Indeed, the brain-wide expression of mGlu4 suggests that modulation of a broad range of brain regions might be expected. This is exactly the motivation to explore the interaction between mGlu4 modulation and impulsivity, which should filter those areas in which mGlu4 might modulate impulsive traits.

We have now addressed this issue in the text of the Results sections (Lines 331-336).

– Authors report they recorded LFPs, but do not show these data in frequency bands; please show these. The unit recording component of this collection of studies is particularly intriguing and represents one of the more novel aspects of the work. One question that arises, however, stems from the fMRI experiments. Were the 3 mice in this study LI or HI? What is the behavioral context for these neuronal responses? If these mice were LI, what neuronal responses might be found in HI, or vice versa? If they were average impulsivity mice, how might these observations extend to HI or LI subjects? Please clarify.

We thank the reviewer for suggesting for this additional, informative analysis of the electrophysiological data regarding HI/LI traits and LFPs.

Regarding HI/LI traits, these animals were averagely impulsive mice, according to our trait criteria, when they were in their naïve state with no electrodes implanted (Figure 6 —figure supplement 7A); their behavior changed during electrophysiological recordings probably due to the presence of the electrodes (Figure 6 —figure supplement 7B). Please note that this cohort responded to mGlu4 PAM treatment as expected, with increased impulsivity (Figure 6 —figure supplement 7B), supporting the validity of our mGlu4 PAM electrophysiological data. Unfortunately, because our working conditions were restricted due to the Covid-19 pandemic conditions we were unable to increase the number of animals in this analysis and, therefore, could not split these animals into HI/LI cohorts. This does not undermine the main findings and conclusion, in our opinion.

Importantly, we have taken up the suggestion to thoroughly analyze the dynamics of LFPs in our task and now include a full LFP analysis for both the STN and SNr networks under baseline conditions and with mGlu4 PAM treatment (Figure 2F-G, Figure 2 —figure supplement 3; Figure 5D-E, Figure 5 —figure supplement 3).

This new analysis led to two important observations. First, the overall power of LFP in the STN was stronger during periods of immobility periods than it was during task periods and bound to several behavioral parameters in GNG task. This picture recapitulates the decoupling of STN from global networks in HI animals (Figure 1Ei). Second, we also noticed significant modulation of the LFP β band during impulsive action (precue poke), which is particularly visible in the SNr and consistent with the idea of uncoupling of the SNr from its network and loss of behavioral inhibition during impulsive action.

– A similar question to that above extends to the optogenetic experiments. The performance of the animals in the control (no laser) conditions appears to be quantitatively similar to the HI mice in the MRI experiment. Were mice selected for HI phenotype? If they were a random group, this raises the question of reproducibility of the LI vs. HI phenotype.

We thank the reviewer for suggesting this additional analysis on HI/LI optogenetic animals and additional data supporting the reproducibility of the HI/LI phenotype.

Technically, 3R policies for reducing animal numbers precluded prescreening larger cohorts to increase the size of the HI and LI animal groups for these experiments. Instead, we split the individual cohorts into HI and LI animals (Figure 6 —figure supplement 7A, colored). As for the electrophysiological cohort, we could not split the animals in the optogenetics experiments into two cohorts because of the small sample number (Figure 6 —figure supplement 7A, grey).

As suggested by this reviewer, we assessed the overall reproducibility of the LI vs. HI behavior, and thus phenotypic characterization, across experimental cohorts (Figure 6 —figure supplement 7A, colored). Overall, the behavior was consistent across experiments and, in cases where splits were affordable, i.e. Figure 1 —figure supplement 1A, Figure 4 —figure supplement 1A, Figure 6 —figure supplement 1, Figure 6 —figure supplement 3. We are therefore convinced our HI/LI phenotype to be robust.

– Only a single dose of the mGlu4 PAM was administered, which limits interpretation of the findings. In addition, the drug was administered systemically, where it could interact with mGlu4 receptors in a number of brain regions to produce its effects. Site-specific microinjections would have been more informative. Authors are encouraged to address this limitation in the Discussion.

We agree that site-specific microinjection of the mGlu4 PAM might have been more informative than systemic administration.

A rationale for systemic administration has now been included in the Limitations of the study section, while shRNAi-mediated site-specific silencing of mGlu4 in the STN does address the question of the site-specific action of this receptor.

Although site-specific administration of mGlu4 PAM might have corroborated our findings with shRNAi-mediated site-specific silencing, they would not have allowed us to conclude anything about the possible importance of other brain regions that express mGlu4. The same argument applies to the initial fMRI, which placed the interaction of the STN with impulsivity in the context of the whole brain. We believe that such holistic embedding of our STN x mGlu4 findings is an asset of the paper that adds significantly to our understanding of the potential of mGlu4 as a therapeutic target.

As suggested, we have conveyed this point more clearly now in the Discussion (Lines 532 and 557).

– Related to the pharmacology aspect and clarifications in the Discussion: the conceptual novelty may be a bit overstated as others have shown before that there is an interaction between D2 receptor antagonism and allosteric modulation of mGluR4 in impulsivity (using traditional pharmacology and a 5-Choice Serial Reaction Time Task). Motor and choice impulsivity (akin to what the authors refer to as trait impulsivity) has been previously dissociated via the mGluR4 receptor (Isherwood et al,. 2017 Neuropharm). So there is precedent for its potential involvement in impulsivity albeit using mostly classic systemic administration methods. Indeed, the role of mGluR4 may likely be via D2R, which is an older story. It is recommended that authors incorporate a consideration of this into their Intro and/or Discussion.

We are aware that mGlu4 has been shown to modulate impulsivity possibly by an interaction with D2R. We do not claim that our finding of a role for mGlu4 in impulsivity is novel.

Rather, we provide mechanistic insight into the dynamics of how mGlu4 PAM modulates brain circuitry at the level of the STN, which is novel in scope and depth. Additionally, we put these new findings into the perspective of a global survey of brain networks and candidate structures, as well as previously unknown local neuronal dynamics, underlying impulsive traits and their modulation by mGlu4. Furthermore, we provide first data indicating that mGlu4 is causally involved in the modulation of impulsive traits. This is an important finding considering the sparse literature on neuronal mechanisms underlying trait impulsivity.

To address these points in the text, we extended the section that contextualizes the novelty of our findings with respect to the literature and modified the Discussion to explain this novelty (Lines 532-557), as well as in the final paragraph of the Discussion (Lines 576-590).

– The authors indicate that they intentionally sought to examine individual differences in these impulsivity traits in an isogenic strain in order to study them in the "same genetic and neuroanatomical context." This statement is somewhat perplexing as the largest source of individual differences in brain mechanisms recruited during go/no-go performance in human subjects is genetic (Anokhin et al., Int J Psychophysiol. 2017 May;115:112-124), as it is for impulsivity, more broadly. Thus, the mechanisms underlying inter-individual variation in impulsive behavior in the isogentic strain is explicitly not the same as the major driving factor influencing this in humans, potentially compromising the translational value of the findings. More discussion of this point is warranted, as are the rationale for, and limitations of, using an only male subject pool.

We agree that a better explanation of our design and interpretation is needed.

Indeed, we now describe how mGlu4 may modulate impulsivity and impulsive traits in an isogenic setting. Screening of various isogenic mouse lines in a behavioral assay can identify genetic loci for behavioral control, as illustrated by a previous study (Loos et al., 2009) that uncovered a correlation between several mPFC genes and impulsivity. We agree with the reviewer that underlying inter-individual variation in impulsive behavior in the isogenic strain may overestimate its role as a major driving factor on impulsivity. We now acknowledge this in the Limitation of the study section (Lines 592-598). Nevertheless, trait differences within an isogenic strain can be used to study environmentally determined (acquired) and/or epigenetic factors that play a role in behavioral expression. Our focus on mGlu4 was based on human GWAS studies and observations indirectly linking the *GRM4* gene to impulsivity, which provided the rationale for our investigations (Supplementary Table 1). Our experiments, therefore, did not screen for new genes correlating with impulsivity or underlying trait impulsivity but focused on mGlu4 and explored its role in impulsivity and impulsive traits. The experiments were performed on animals with an isogenic background to exclude putative confounding genetic factors, and to assess the role of acquired impulsive traits and putative inter-individual differences based on epigenetics.

Moreover, surveys of the World Health Organization have shown that there is a significant gender bias towards male individuals in the prevalence of ADHD (Fayyad et al., 2017; Zhang et al., 2021). The use of male subjects only is not unique to our work, as sex specific studies have been performed in both rodent and human subjects by others (Bellés et al., 2021; Heinrich et al., 2017; Jung et al., 2020; Sorokina et al., 2018; Zapata and Lupica, 2021). Of course, we do not wish to downplay the importance of female trait impulsivity and it remains an important topic for future studies if our findings diverge with gender.

– There appears to be some overflow of virus expression into peri-STN regions such as the zona incerta, which the authors attribute to filling of neurites and/or projections. Higher magnification images showing that expression in these regions is confined to fibers and not cell bodies are needed to demonstrate this. This is especially important when stimulating with red light since it is capable of penetrating further into tissue than blue light.

We thank this Reviewer for the suggestion to communicate better the specificity of the stereotaxic/optogenetic targeting.

We noticed that all our histological images appeared oversaturated after CMYK conversion so we have now adapted this throughout the manuscript wherever necessary (Figure 3 —figure supplement 1, Figure 6 —figure supplement 2). These images now represent better our automated quantification (Figure 3 —figure supplement 1B top, Figure 6 —figure supplement 2B, top).

First, injections across optogenetic and shRNAi cohorts all consistently showed fluorescence was enriched in the STN (Figure 3 —figure supplement 1B top, Figure 6 —figure supplement 2B, top), in addition to STN cell body expression (Figure 3 —figure supplement 1B, bottom, Figure 6 —figure supplement 2B, bottom). Note that despite its lower detectability in the Arch cohort the primary site of viral infection and manipulation was STN. For the optogenetic cohorts, targeting of the STN was further supported by the axonal (but not cell body) labelling of known STN targets in the SNr and ZI target regions across all cohorts, including Arch injected animals (Figure 3 —figure supplement 1B, bottom). To support this further, we now provide higher magnification images of the targeted regions, clearly showing infected cell bodies in the STN expressing GFP (Figure 3 —figure supplement 1Bi), ChR2 (Figure 3 —figure supplement 1Bii) and Arch (Figure 3 —figure supplement 1Biii) and transgene transport to their projection targets in ZI and SNr. Taken together, we believe that viral infection primarily targeted the STN in all our manipulations.

Second, the scattered, off-target infection of some cell bodies in the ZI, mainly observed in the optogenetic cohort (Figure 3 —figure supplement 1B, bottom), lies dorsal to and flanks the tip of the optical fiber and therefore would not be illuminated by back-scattered light, which is not effective enough to cause activation/silencing in the optogenetic experiments. Note that the electrophysiological data that demonstrate that STN neurons are effectively targeted by the viral vectors and modulated by light in both ChR2 and Arch (Figure 3 —figure supplement 2). Therefore, robust silencing and activation can be expected only in the STN, which is the primary site of infection and expression and the only infected region directly illuminated by the fiber tip (Figure 3 —figure supplement 2A).

Collectively, we believe our optogenetic and shRNAi viral manipulations are largely directed at and specific for STN.

– The mGlu4 hybridization signal is not exceptionally strong, nor is it concentrated in cell soma of vGluT positive neurons. Only a few red pixels can be observed, and some message signal appears to be extracellular or on fibers that are not labeled. A more convincing demonstration of mGlu4 expression in vGluT positive neurons is needed.

We thank this Reviewer for the suggestion to characterize mGlu4 expression more deeply.

The STN is a known site of mGlu4 expression and it is largely glutamatergic. Indeed, new automated quantification reveals that there is an enrichment of mGlu4 signal in perisomatic areas of cells over the general neuropil. Moreover, the mGlu4 signal in STN cell bodies is correlated with trait impulsivity (Figure 6 —figure supplement 1Bi). Importantly, this quantification also confirmed previous literature in that mGlu4 is enriched in STN VGlut+ cells, and in particular in HI animals (Figure 6 —figure supplement 1Bii).

We hope that the additional analyses convincingly demonstrate that mGlu4 is expressed in vGluT-positive neurons, so supporting our claim that mGlu4 in glutamatergic neurons of the STN underlies our observations. We do not wish to rule out, however, the possibility that inhibitory cells may also contribute.

– Both the behavioral data from the mGluR4 PAM study and the mGluR4 knockdown study involved separate ANOVA in the HI and LI groups. This is not justified. An omnibus ANOVA with impulsivity trait group as a factor should be conducted, and only in the case of a relevant interaction should the groups be analyzed separately.

We thank this Reviewer for this excellent suggestion.

We now used HI/LI percentile or median split as a factor for our statistical analysis, as discussed elsewhere (Iacobucci et al., 2015), where group size/experimental power allowed (Figure 1 —figure supplement 1A, Figure 4 —figure supplement 1A, Figure 6 —figure supplement 1, Figure 6 —figure supplement 3 ; Figure 6 —figure supplement 7A).

We contrasted these splits for functional pharmacological/genetic perturbations with traits as a factor using 3-way ANOVAS (Figure 4C, Figure 6E). A strong trend for an interaction in the mGlu4 PAM and significance in the mirE experiment was seen in this test. Within the experimental limitations (group sizes) we are confident that overall there seems a bias for a trait dependency of these effects.

– In order to fully understand the effects of mGluR4 knockdown on behavior in HI and LI mice, molecular data on mGluR4 protein in HI and LI mice exposed to the active or control virus would be very helpful.

Quantification of mGlu4 protein in the STN proved to be impractical for two reasons. First and foremost, quantification of western blots and immunohistochemistry of STN tissue was technically very difficult because of the low level of mGlu4 protein in this tissue and the poor sensitivity of the antibody. Secondly, even if we were able to detect mGlu4 protein in the STN, it would be difficult to distinguish between mGlu4 at SN or GP synapses, which receive projections from mGlu4-positive areas other than STN. This might be possible, in principle, by co-labelling of virally injected terminals, however, immunohistochemistry was too insensitive in our hands. Although this type of analysis would be very valuable, we feel it is beyond our abilities at this point.

– A nontrivial 11 AAV::Arch animals were excluded for non-performance/technical issues. Can authors provide more information about this? Non-performance is very different than technical issues as an exclusion criterion. Please justify.

For added transparency we have now explained the reasons for including and excluding animals based on technical criteria, performance and histology.

This now better described in the Methods section.

Technical exclusions were in the case of software malfunction across all cohorts, excessive coiling of the laser fibers or the loss of at least one fiber during the task for the optogenetic cohorts, and malaise due to gavaging for the mGlu4 vehicle/PAM cohorts.

The comparatively large number of exclusions from the optogenetic Arch cohort arose from defective fiber adaptors/twist compensation in a subset of our setups, which we noticed too late during the experiments on this cohort but which are visible in the videos.

Non-performance exclusions were all cases of animals that failed to meet a threshold of 80% for correct Go and 45% for FA at baseline in three consecutive sessions (post-surgery, if applicable).

Experimental outliers refer to data points excluded for statistical power after stratification by Grubbs test outlier removal for individual parameters using conservative settings (α = 0.0001).

– For optogenetics studies, viral vectors used incorporated a pan-neuronal promoter (hSyn). Is there a reason why the authors did not choose a more selective cell-type specific promoter, say for excitatory neurons? Please clarify.

In this exploratory study, we wanted to establish whether there was any role of the STN and mGlu4 in modulation of impulsive traits; therefore, we used a pan-neuronal promoter. A detailed study dissecting the role of individual neurons and their projections might be well suited for a follow up study, accompanied and complementing a deeper analyses of specific fMRI impulsivity subnets (see Main 2, discussion about nodes vs edges).

– Can the authors cite prior studies by other groups that have used similar behavioral splits based on distance travelled in the GNG task, licks, etc.?

The choice of parameters for our fMRI study were based on behavioral aspects that accompany impulsivity, such as hyperactivity (Majdak et al., 2016) and behaviors related to reward seeking (Dalley et al., 2011). Thus, we examined the video recordings of GNG tasks for the parameters of distance traveled (analogous to open field) and licks (a method used in head-fixed GNG tasks). This allowed us to detect the involvement of various nodes in different aspects of impulsive behavior, which is a novel aspect of this study. Statistical splits, including but not restricted to median splits *per se* in impulsivity studies, have been used by others, however, those studies focused only on the impulsivity-related parameters of the assessed task (Caprioli et al., 2014; Isherwood et al., 2017; Moloney et al., 2019; Sánchez-Kuhn et al., 2017; Ucha et al., 2019). Thus, in our study we consider these splits on behaviors orthogonal to impulsivity parameters, which are a novel and informative feature of our study.